# Weisfeiler and Lehman Go Cellular: CW Networks

**Cristian Bodnar**[*]
University of Cambridge
cb2015@cam.ac.uk

**Fabrizio Frasca**[*]
Imperial College London & Twitter
ffrasca@twitter.com

**Nina Otter**
UCLA
otter@math.ucla.edu

**Yu Guang Wang**
MPI-MIS, SJTU & UNSW
yuguang.wang@unsw.edu.au

**Pietro Liò**
University of Cambridge
pl219@cam.ac.uk

**Guido Montúfar**
MPI-MIS & UCLA
montufar@math.ucla.edu

**Michael Bronstein**
Imperial College London & Twitter
mbronstein@twitter.com

## Abstract

Graph Neural Networks (GNNs) are limited in their expressive power, struggle with long-range interactions and lack a principled way to model higher-order structures. These problems can be attributed to the strong coupling between the computational graph and the input graph structure. The recently proposed Message Passing Simplicial Networks naturally decouple these elements by performing message passing on the clique complex of the graph. Nevertheless, these models can be severely constrained by the rigid combinatorial structure of Simplicial Complexes (SCs). In this work, we extend recent theoretical results on SCs to regular Cell Complexes, topological objects that flexibly subsume SCs and graphs. We show that this generalisation provides a powerful set of graph "lifting" transformations, each leading to a unique hierarchical message passing procedure. The resulting methods, which we collectively call CW Networks (CWNs), are strictly more powerful than the WL test and not less powerful than the 3-WL test. In particular, we demonstrate the effectiveness of one such scheme, based on rings, when applied to molecular graph problems. The proposed architecture benefits from provably larger expressivity than commonly used GNNs, principled modelling of higher-order signals and from compressing the distances between nodes. We demonstrate that our model achieves state-of-the-art results on a variety of molecular datasets.

## 1 Introduction

The operations performed by message passing Graph Neural Networks (GNNs) emulate the structure of the input graph. While this property has clear computational advantages, it brings with it a series of fundamental limitations. As observed by Xu et al. [74] and Morris et al. [55] the local neighbourhood aggregations used by GNNs are at most as powerful as the Weisfeiler-Lehman (WL) test [71] in distinguishing non-isomorphic graphs. Therefore, GNNs fail to detect certain higer-order meso-scale structures such as cliques or (induced) cycles [2, 15], which are particularly important in applications dealing with social and biological networks or molecular graphs. At the same time, many such layers have to be stacked to make long-range interactions in the graph possible. Besides the computational burden this incurs, deep GNNs typically come with additional problems such as over-smoothing [51] and over-squashing [1] of the node representations.

---

[*]Authors contributed equally.

To address these problems, we propose a novel message passing procedure based on (regular) cell complexes, also known as CW complexes[2], topological objects that form the building block of algebraic topology [38]. When paired with a theoretically-justified "lifting" transformation augmenting the graph with higher-dimensional constructs called "cells", our method results in a multi-dimensional and hierarchical message passing procedure over the input graph. Our approach generalises and subsumes the recently proposed Message Passing Simplicial Networks (MPSNs) [8], which operate on simplicial complexes (SCs), topological generalisations of graphs. However, SCs have a rigid combinatorial structure that significantly limits the range of lifting transformations one could use to meaningfully modulate the message passing procedure. In contrast, we show that cell complexes, which in turn generalise simplicial complexes and come with additional flexibility, allow one to construct new and better ways of decoupling the input and computational graphs.

**Main Contributions** To summarise, we propose a message passing scheme operating on regular cell complexes. We call this family of models CW Networks (CWNs) and study their expressive power using a cellular version of the WL test. We show that for an entire class of "lifting" transformations CWNs are at least as powerful as the WL test. Furthermore, we prove that for some of the maps in this class, CWNs can be strictly more powerful than WL, Simplicial WL (SWL) and also not less powerful than 3-WL. We also express the fundamental symmetries of these models and show how they can be seen as generalised convolutional operators on cell complexes. Experimentally, we focus our attention on a particular "lifting" map based on induced cycles. When applied to molecular graphs, it leads to an intuitive hierarchical message passing procedure involving the atoms, the bonds between them and the chemical rings of the molecules. We demonstrate that this provably powerful approach obtains state-of-the-art results on popular large-scale molecular graph datasets and other related tasks. To the best of our knowledge, this is the first work proposing a cell complex representation for molecules. Our code is available at `https://github.com/twitter-research/cwn`.

## 2 Background

**Definition 1** (Hansen and Ghrist [36])**.** *A **regular cell complex** (Figure 1) is a topological space $X$ together with a partition $\{X_\sigma\}_{\sigma \in P_X}$ of subspaces $X_\sigma$ of $X$ called **cells**, and such that*

*1. For each $x \in X$ there exists an open neighborhood of $x$ that intersects finitely many cells.*

*2. For all $\sigma, \tau$ we have that $X_\tau \cap \overline{X_\sigma} \neq \emptyset$ iff $X_\tau \subseteq \overline{X_\sigma}$, where $\overline{X_\sigma}$ is the closure of a cell.*

*3. Every cell is homeomorphic to $\mathbb{R}^n$ for some $n$.*

*4. (Regularity) For every $\sigma \in P_X$ there is a homeomorphism $\phi$ of a closed ball in $\mathbb{R}^{n_\sigma}$ to $\overline{X_\sigma}$ such that the restriction of $\phi$ to the interior of the ball is a homeomorphism onto $X_\sigma$.*

We note that by condition (2) the indexing set $P_X$ has a poset structure $\tau \leq \sigma \Leftrightarrow X_\tau \subseteq \overline{X_\sigma}$, while condition (4) guarantees that this poset structure encodes all the topological information about $X$. Thus, we can identify a regular cell complex $X$ with this poset, called **face poset** of $X$. We also use $\tau < \sigma$ for the strict version of this partial order.

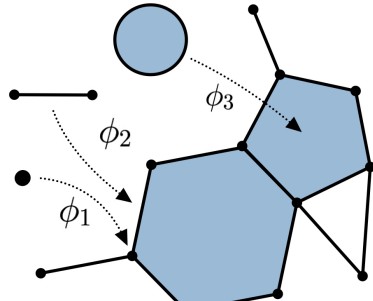

Figure 1: A cell complex $X$ and the corresponding homeomorphisms to the closed balls for three cells of different dimensions in the complex.

Intuitively, one constructs a cell complex through a hierarchical gluing procedure. One starts with a set of vertices (0-cells). Then edges (1-cells) are attached to these by gluing the endpoints of closed line segments to them. We have now only described a (multi) graph. However, one can generalise this even further by taking a two-dimensional closed disk and glue its boundary (i.e. a circle) to any simple cycle in the (multi) graph previously built as in Figure 2. While we are generally not concerned with dimensions above two, this can be further generalised by gluing the boundary of $n$-dimensional balls to certain $(n-1)-$cells in the complex.

Consider the examples in Figure 3. The shown sphere is a cell complex obtained from two 0-cells (i.e. vertices), to which two 1-cells (i.e. edges), which form the equator, were attached. The boundary

---

[2]We use these terms interchangeably. For the latter, the C stands for "closure-finite", and the W for "weak" topology. The term was coined by Whitehead [72].

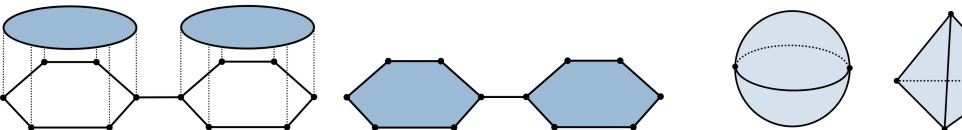

Figure 2: Closed two-dimensional disks are glued to the boundary of the rings present in the graph (left). The result is a 2D regular cell complex (right).

Figure 3: A sphere and an empty tetrahedron. The latter is also a simplicial complex.

of two 2-dimensional disks (i.e. the two hemispheres) were glued to the equator to form a sphere. The second example is a tetrahedron with empty interior. It is a particular type of cell complex called a *simplicial complex* (SC). The only 2-cells it allows are triangle-shaped. More generally, the $n$-dimensional cells of SCs are $n$-simplices, which makes them slightly more rigid structures.

**Definition 2.** *The **k-skeleton** of a cell complex $X$, denoted $X^{(k)}$, is the subcomplex of $X$ consisting of cells of dimension at most $k$.*

This definition is useful for referring for certain parts of the complex. For instance, $X^{(0)}$ contains the vertices in the complex, while $X^{(1)}$ contains the vertices *and* the edges (i.e. the underlying graph).

The combinatorial structure of the complex can be more compactly described by an incidence relation we call the *boundary relation*, whose reflexive and transitive closure gives the partial order defined above. The boundary relation describes what cells are on the boundary of other cells. For instance, the edges of the sphere in Figure 3 are on the boundary of the 2-cells forming the two hemispheres.

**Definition 3.** *We have the **boundary relation** $\sigma \prec \tau$ iff $\sigma < \tau$ and there is no cell $\delta$ such that $\sigma < \delta < \tau$.*

We can use this to define the four types of (local) adjacencies present in cell complexes. These adjacencies will be the fundamental building block of our message passing procedure. To explain these in more familiar terms, for each adjacency, we exemplify how it shows up in graphs.

**Definition 4** (Cell complex adjacencies). *For a cell complex $X$ and a cell $\sigma \in P_X$, we define:*

1. *The boundary adjacent cells $\mathcal{B}(\sigma) = \{\tau \mid \tau \prec \sigma\}$. These are the lower-dimensional cells on the boundary of $\sigma$. For instance, the boundary cells of an edge are its vertices.*

2. *The co-boundary adjacent cell $\mathcal{C}(\sigma) = \{\tau \mid \sigma \prec \tau\}$. These are the higher-dimensional cells with $\sigma$ on their boundary. For instance, the co-boundary cells of a vertex are the edges it is part of.*

3. *The lower adjacent cells $\mathcal{N}_\downarrow(\sigma) = \{\tau \mid \exists \delta \text{ such that } \delta \prec \sigma \text{ and } \delta \prec \tau\}$. These are the cells of the same dimension as $\sigma$ that share a lower dimensional cell on their boundary. The line graph adjacencies between the edges are a classic example of this.*

4. *The upper adjacent cells $\mathcal{N}_\uparrow(\sigma) = \{\tau \mid \exists \delta \text{ such that } \sigma \prec \delta \text{ and } \tau \prec \delta\}$. These are the cells of the same dimension as $\sigma$ that are on the boundary of the same higher-dimensional cell as $\sigma$. The typical graph adjacencies between vertices are the canonical example here.*

## 3 Cellular Weisfeiler Lehman

**Overview** The results in this section show how one can transform graphs into higher-dimensional cell complexes in such a way that performing colour refinement on the resulting cell complexes makes it easier to test their isomorphism. The message passing model from Section 4 will take advantage of these theoretical results. All proofs can be found in Appendix A.

**Definition 5.** *Let $c$ be a colouring of the cells in a complex $X$ with $c_\sigma$ denoting the colour assigned to cell $\sigma \in P_X$. Define $\mathcal{B}(\sigma, \tau) := \mathcal{B}(\sigma) \cap \mathcal{B}(\tau)$ and $\mathcal{C}(\sigma, \tau) := \mathcal{C}(\sigma) \cap \mathcal{C}(\tau)$. We define the following multi-sets of colours:*

1. *The colours of the boundary cells of $\sigma$: $c_\mathcal{B}(\sigma) = \{\!\{c_\tau \mid \tau \in \mathcal{B}(\sigma)\}\!\}$.*

2. *The colours of the co-boundary cells of $\sigma$: $c_\mathcal{C}(\sigma) = \{\!\{c_\tau \mid \tau \in \mathcal{C}(\sigma)\}\!\}$.*

3. *The lower adjacent colours of $\sigma$: $c_\downarrow(\sigma) = \{\!\{(c_\tau, c_\delta) \mid \tau \in \mathcal{N}_\downarrow(\sigma) \text{ and } \delta \in \mathcal{B}(\sigma, \tau)\}\!\}$.*

4. *The upper adjacent colours of $\sigma$: $c_\uparrow(\sigma) = \{\!\{(c_\tau, c_\delta) \mid \tau \in \mathcal{N}_\uparrow(\sigma) \text{ and } \delta \in \mathcal{C}(\sigma, \tau)\}\!\}$.*

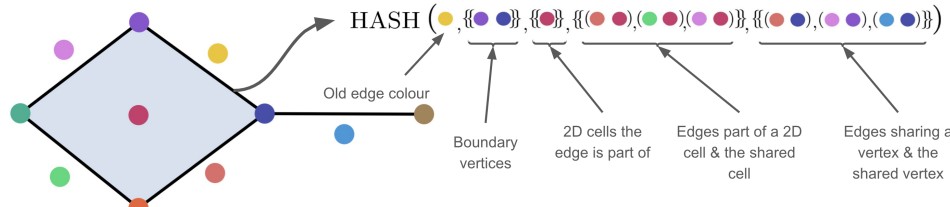

Figure 4: The CWL colouring procedure for the yellow edge of the cell complex. All cells have been assigned unique colours to aid the visualisation of the adjacencies. Note that the yellow edge aggregates long-range information from the light green edge.

Note that unlike in graphs and simplicial complexes, the sets $\mathcal{B}(\sigma, \tau)$ and $\mathcal{C}(\sigma, \tau)$ can have more than one element. For instance, two (closed) 2-cells might intersect in more than one edge (e.g. the two hemispheres in Figure 3), and conversely, two edges might be on the boundary of the same two 2-cells. This illustrates the more flexible combinatorial structure of cell complexes.

**Cellular WL (CWL)** We consider CWL, a colour refinement scheme for cell complexes that generalises the Simplicial WL [8] and WL [71] tests. We use $c_\sigma^t$ to refer to the colour assigned by CWL to cell $\sigma$ at iteration $t$ of the algorithm. When the input is a simplicial complex, this recovers the SWL algorithm. A step of the algorithm is graphically depicted in Figure 4 for a single cell.

1. Given a regular cell complex $X$, all the cells $\sigma$ are initialised with the same colour.
2. Given the colour $c_\sigma^t$ of cell $\sigma$ at iteration $t$, we compute the colour of cell $\sigma$ at the next iteration $c_\sigma^{t+1}$ by injectively mapping the multi-sets of colours belonging to the adjacent cells of $\sigma$ using a perfect HASH function: $c_\sigma^{t+1} = \text{HASH}\big(c_\sigma^t, c_{\mathcal{B}}^t(\sigma), c_{\mathcal{C}}^t(\sigma), c_\downarrow^t(\sigma), c_\uparrow^t(\sigma)\big)$.
3. The algorithm stops when a stable colouring is reached. Two cell complexes are considered non-isomorphic if their colour histograms are different. Otherwise, the test is inconclusive.

First, we state the following theorem from Bodnar et al. [8] involving SWL and simplicial complexes. This theorem shows that on simplicial complexes, certain adjacencies can be pruned without affecting the non-isomorphic SCs that can be distinguished. This has important computational implications.

**Theorem 6.** *SWL without coboundary and lower-adjacencies has the same expressive power in distinguishing non-isomorphic simplicial complexes as SWL with the complete set of adjacencies.*

It is not immediately clear whether an equivalent theorem would also hold for cell complexes. This is because cells, unlike simplices, can have widely different shapes and, as described above, the adjacencies between them take more complicated forms. Nevertheless, we show that a positive result can be obtained.

**Theorem 7.** *CWL without coboundary and lower-adjacencies has the same expressive power in distinguishing non-isomorphic cell complexes as CWL with the complete set of adjacencies.*

We note this does not mean that the removed adjacencies are completely redundant in practice. Even if they are not needed from a (theoretical) colour refinement perspective, they might still include important inductive biases that make them suitable for certain tasks.

We are now interested in examining various procedures for mapping, or "lifting", graphs into the space of regular cell complexes. Such a procedure can be used to test the isomorphism of two graphs by performing colour refinement on the cell complexes they are mapped to. The hope is that CWL applied to these cell complexes is more powerful than WL applied to the initial graphs. We will later show that for a wide range of transformations, this is indeed the case. We start by rigorously defining what we mean by a "lifting".

**Definition 8.** *A **cellular lifting map** is a function $f : \mathcal{G} \to \mathcal{X}$ from the space of graphs $\mathcal{G}$ to the space of regular cell complexes $\mathcal{X}$ with the property that two graphs $G_1, G_2$ are isomorphic iff the cell complexes $f(G_1), f(G_2)$ are isomorphic.*

This property ensures that testing the isomorphism of the two cell complexes is equivalent to testing the isomorphism in the input graphs. This would not be the case if two non-isomoprhic graphs were mapped to the same cell complex.

**Example 9.** *It can be verified that the function mapping each graph to its clique complex (i.e. every $(k+1)$-clique in the graph becomes a $k$-simplex) is a cellular lifting map.*

The clique complex lifting map from Example 9 has been used by Bodnar et al. [8] to show that SWL is strictly more powerful than WL. We restate this result:

**Theorem 10.** *SWL with clique complex lifting is strictly more powerful than WL.*

A natural question is what other lifting transformations make CWL strictly more powerful than WL? We first describe a space of lifting transformations that make CWL at least as powerful as WL.

**Definition 11.** *A lifting map is **skeleton-preserving** if for any graph $G$, the 1-skeleton of $f(G)$ and $G$ are isomorphic as (multi) graphs.*

Intuitively, skeleton-preserving liftings ensure that the additional structure added by the lifting map comes from attaching cells of dimension at least two to the graph. These mappings keep the 0-cells and 1-cells intact and are, therefore, restricted from making modifications to the input graph structure. An important remark is that for simplicial complexes, attaching simplices based on cliques present in the graph is the only possible skeleton preserving transformation. Once again, this illustrates the limitations of simplicial complexes for adding useful higher-dimensional structures to the graph.

**Example 12.** *The function from Example 9 is also skeleton-preserving because the 1-skeleton of the clique complex of a graph is trivially isomorphic to the graph. A lifting function mapping each graph to a multi-graph where each edge is doubled by a parallel edge is not skeleton-preserving (Figure 5).*

We now show that all the maps in the skeleton-preserving class have the following desirable property:

**Theorem 13.** *Let $f$ be a skeleton-preserving lifting map. Then CWL($f$) (i.e. CWL using lifting $f$) is at least as powerful as WL in distinguishing non-isomorphic graphs.*

To prove that some of these make CWL strictly more powerful than WL, it is sufficient to find a pair of graphs that cannot be distinguished by WL, but can be distinguished by CWL. The following result gives examples of such maps.

Figure 5: A graph, its clique complex and the graph with duplicated edges. The first map is skeleton-preserving, while the second is not.

**Definition 14.** *Let $k$-CL, $k$-IC, $k$-C be the lifting maps attaching cells to all the cliques, induced cycles and simple cycles, respectively, of size at most $k$.*

**Corollary 15.** *For all $k \geq 3$, CWL($k$-CL), CWL($k$-IC) and CWL($k$-C) are strictly more powerful than WL.*

We note that this is not a complete list. For instance, the result can also be extended to combinations of the above or other transformations. We can also relate CWL to the higher-order 3-WL test.

**Theorem 16.** *There exists a pair of graphs indistinguishable by 3-WL but distinguishable by CWL($k$-CL) with $k \geq 4$, CWL($k$-IC) with $k \geq 4$ and CWL($k$-C) with $k \geq 8$.*

Finally, we conclude this section by showing how CWL can achieve a superior expressive power compared to SWL. This result is proven by Corollary 31 in the Appendix.

**Theorem 17.** *Let $k$-CL $\cup$ $k$-IC and $k$-CL $\cup$ $k$-C denote combined liftings attaching cells to the union of the specified substructures. CWL($k_1$-CL $\cup$ $k_2$-IC) and CWL($k_1$-CL $\cup$ $k_2$-C) are strictly more powerful than SWL($k_1$-CL) for all $k_2 \geq 5$.*

## 4 Molecular Message Passing with CW Networks

We now describe CW Networks with an applied focus on molecular graphs to ground the discussion. Therefore, from now on we assume the use of the skeleton-preserving lifting transformation that attaches 2-cells to all the induced cycles (i.e. chordless cycles) in the graph as in Figure 2. This leads to a message passing procedure involving atoms (vertices / 0-cells), the bonds between atoms (edges / 1-cells) and chemical rings (induced cycles / 2-cells). Additionally, in virtue of Theorem 7, we consider only the boundary and upper adjacencies between these cells without sacrificing the expressive power. The equations for the other adjacencies, which we do not use, can be found in Appendix A. We note however, that the theoretical results in this section are general and not particular to these specific choices of adjacencies and lifting transformation.

**Molecular Message Passing** The cells in our CW Network receive two types of messages:

$$m_{\mathcal{B}}^{t+1}(\sigma) = \text{AGG}_{\tau \in \mathcal{B}(\sigma)}\Big(M_{\mathcal{B}}\big(h_\sigma^t, h_\tau^t\big)\Big) \qquad m_\uparrow^{t+1}(\sigma) = \text{AGG}_{\tau \in \mathcal{N}_\uparrow(\sigma), \delta \in \mathcal{C}(\sigma, \tau)}\Big(M_\uparrow\big(h_\sigma^t, h_\tau^t, h_\delta^t\big)\Big).$$

The first specifies messages from atoms to bonds and from bonds to rings. The second type of message, specifies messages between atoms connected by a bond and messages between bonds that are part of the same ring (Figure 6). Note that for the second type of adjacency, when two atoms communicate, we include the features of the bond between them. Similarly, when two bonds communicate, we include the features of the ring they communicate through. The update operation takes into account these two types of incoming messages and updates the features of the cells:

$$h_\sigma^{t+1} = U\Big(h_\sigma^t, m_{\mathcal{B}}^t(\sigma), m_\uparrow^{t+1}(\sigma)\Big). \tag{1}$$

To obtain a global embedding for a cell complex $X$ from a model with $L$ layers, the readout function takes as input the separate multi-sets of features corresponding to the atoms, bonds and the rings:

$$h_X = \text{READOUT}(\{\!\{h_\sigma^L\}\!\}_{dim(\sigma)=0}, \{\!\{h_\sigma^L\}\!\}_{dim(\sigma)=1}, \{\!\{h_\sigma^L\}\!\}_{dim(\sigma)=2}). \tag{2}$$

**Expressivity** Naturally, the ability of CWNs to distinguish non-isomorphic regular cell complexes is bounded by CWL. Similarly to GNNs and WL, CWNs can also be shown to be as powerful as CWL as long as they are equipped with a sufficient number of layers and the parametric local aggregators they use can learn to be injective. Multiple such multi-set aggregators [20, 74] are known to exist and can be directly employed in our model.

**Theorem 18.** *CW Networks are at most as powerful as CWL. Additionally, when using injective neighbourhood aggregators and a sufficient number of layers, CWNs are as powerful as CWL.*

Corollary 15 states that CWL is strictly more powerful than the standard WL when the lifting procedure attaches 2-cells to induced cycles of size $k \geq 3$. As a consequence of Theorem 18, this result also holds for molecular message passing CWNs equipped with injective aggregators. In practice, $k$ is to be considered as a standard hyperparameter, and its choice can either be driven by validation set performance, or by domain knowledge (if available).

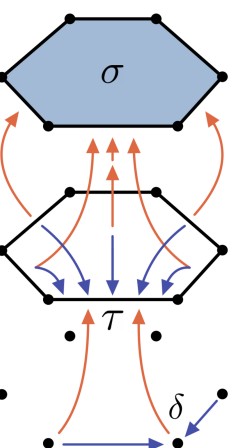

Figure 6: Hierarchical depiction of the message passing procedure. Orange arrows indicate boundary messages received by cells $\sigma$ and $\tau$, while blue ones show upper messages received by cells $\tau$ and $\delta$.

**Symmetries** Given a graph $G$ with adjacency matrix $\boldsymbol{A}$ and feature matrix $\boldsymbol{X}$, a function $f$ is (node) permutation equivariant if $\boldsymbol{P}f(\boldsymbol{A}, \boldsymbol{X}) = f(\boldsymbol{P}\boldsymbol{A}\boldsymbol{P}^T, \boldsymbol{P}\boldsymbol{X})$, for any permutation matrix $\boldsymbol{P}$. GNN layers respect this equation, which ensures they compute the same functions up to a permutation (i.e. relabeling) of the nodes. Similarly, it can be shown that CW Networks are equivariant with respect to permutations of the cells and corresponding permutations of the boundary relations $\sigma \prec \tau$ between cells. We define this notion of equivariance more formally in Appendix C.

**Theorem 19.** *CW Network layers are cell permutation equivariant.*

**Long-Range Interactions** Several graph-related tasks require the ability to capture long-range interactions between nodes. For instance, certain molecular properties depend on atoms placed on the opposite sides of a ring [31, 60]. As a consequence of the coupling between the input and computational graphs, $L$ message passing operations are necessary in GNNs to let a node receive information from an $L$-hops distant node. In contrast, our hierarchical message passing scheme requires *at most* $L$ layers since 2-cells create shortcuts. For example, a constant number of CWN layers (3) is enough to capture dependencies between atoms on the opposite sides of a ring, independently of the ring size. In Section 5.1 we verify this in a controlled scenario. Additional experiments on real world graphs in Section 5.2 confirm that it can achieve state-of-the-art performance with a limited number of layers.

**Anisotropic Filters** Due to the lack of a canonical ordering between neighbours, many common GNNs use symmetric convolutional kernels, resulting in isotropic filters treating neighbours equally. Recent works have proposed to address this limitation by employing additional structural information [6, 10]. CWNs also implicitly achieve this form of anisotropy by integrating information from

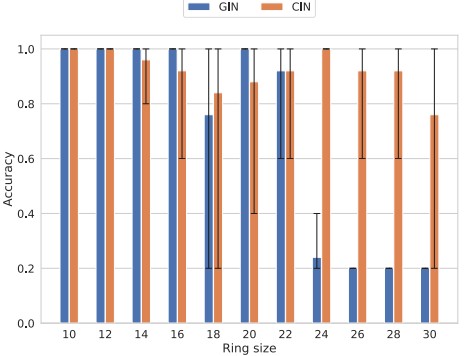

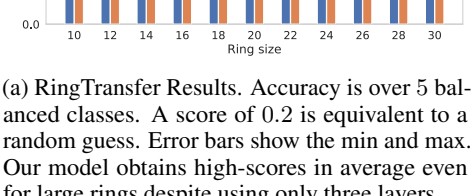

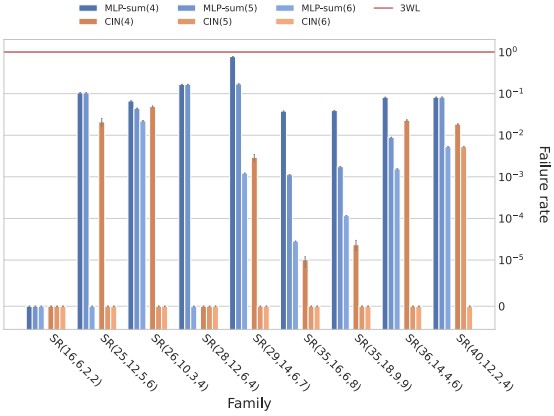

(a) RingTransfer Results. Accuracy is over 5 balanced classes. A score of 0.2 is equivalent to a random guess. Error bars show the min and max. Our model obtains high-scores in average even for large rings despite using only three layers.

(b) Failure rates on the SR isomorphism task, *the smaller the better* (mean and std-error over 5 runs). In parantheses, for each model, the maximum size $k$ of rings lifted to 2-cells.

Figure 7: Results on the RingTransfer and SR synthetic benchmarks.

the higher-order cells and their associated substructures into the message passing procedure. For instance, bond features can learn to encode their membership to a ring and also communicate directly with other bonds present in the ring. Consequently, the messages between atoms connected through these bonds are modulated by the presence of the ring as well as by the presence of other nodes and bonds part of that ring.

**CWNs as Generalised Convolutions** Our message passing scheme can be seen a (non-linear) generalisation of linear diffusion operators on cell complexes. Recent works [13, 25] have introduced convolutional operators on SCs by employing the Hodge Laplacian [63], a generalisation of the graph Laplacian. By leveraging on the cellular Sheaf Laplacian [36], a similar construction can be extended to cell complexes to define cellular convolutional operators. In Appendix D we discuss this approach and show that our cellular message passing scheme subsumes it. This represents a promising avenue for studying CWNs from a spectral perspective, an endeavour we leave for future work.

**Computational Complexity** When considering cells of a constant maximum dimension and boundary size, the computational complexity of the message passing scheme is linear in the size of the input complex. For the molecular applications we are interested in, the average number of rings per molecule is upper bounded by a small constant (e.g. three for MOLHIV), so the size of the complex is approximately the same as the size of the graph. Therefore, in this setting, the computational complexity of the model is similar to that of message passing GNNs. Separately of this, the one-time preprocessing step of computing the lifting of the graphs should also be considered. The $C$ induced cycles in a graph can be listed in $\mathcal{O}\big((|E| + |V|C)$ polylog $|V|\big)$ time [26]. Again, given that $C$ is upper bounded by a small constant for the molecular datasets of interest in this work, the complexity of the lifting procedure is also almost linear in the size of the graph. A more detailed analysis backed up by wall-clock time experiments is given in Appendix B.

## 5 Experiments

In this section we validate the theoretical and empirical properties of our proposed message passing scheme in controlled scenarios as well as in real-world graph classification problems, with a focus on large scale molecular benchmarks. For simplicity, in all experiments we employ a model which stacks CWN layers with local aggregators as in GIN [74]. We name our architecture "Cell Isomorphism Network" (CIN). 0-cells are always endowed with the original node features; higher-dimensional cells are populated in a benchmark specific manner. See Appendix E for details on feature initialisation, message passing and readout operations, hyperparameters, implementation and benchmark statistics.

## 5.1 Synthetic Benchmarks

**CSL** Circular Skip Link dataset was first introduced in [57] and has been recently adopted as a reference benchmark to test the expressivity of GNNs [24]. It consists of 150 4-regular graphs from 10 different isomorphism classes, which we need to predict. Unsolvable by the WL test and message passing approaches [14, 57], we use it to validate the expressive power of CWNs.

Table 1: Classification accuracy on CSL.

| Method | Mean | Min | Max |
|--------|------|-----|-----|
| MP-GNNs | 10.000±0.000 | 10.000 | 10.000 |
| RingGNN | 10.000±0.000 | 10.000 | 10.000 |
| 3WLGNN | 97.800±10.916 | 30.000 | 100.000 |
| CIN (Ours) | 100.000±0.000 | 100.000 | 100.000 |

We follow the same evaluation setting as Dwivedi et al. [24]: 5-fold cross validation procedure and 20 different random weight initialisations. For our model, we set the maximum ring size $k = 8$. In Table 1 we follow the common practice on this dataset and report the mean, minimum and maximum test accuracy obtained by CIN over the 100 runs, along with the results by the baselines presented in Dwivedi et al. [24]. MP-GNNs, that is classic message passing GNNs (GAT [69], MoNet [54], GIN [74], etc.), and RingGNN [14] perform as random guessers. In contrast, our model is able to identify the isomorphism class of each test graph in every run while featuring only a fraction of the computational complexity of 3WLGNN, the best performing reference baseline [24, 53].

**SR** Similarly to Bodnar et al. [8] and [10], we consider Strongly Regular graphs within the same family as hard examples of non-isomorphic graphs we seek to distinguish. Any pair of graphs within the same family cannot provably be distinguished by 3-WL test [8, 10]. We reproduce the same experimental setting of Bodnar et al. [8]. In particular, we consider 9 distinct SR families[3] and run our model untrained on the cell complex lifting of each graph, with $k = 4, 5, 6$. 0-cells (nodes) are initialised with a constant unitary signal, while 1- and 2-cells are initialised with the sum of the contained 0-cells. We additionally run an MLP baseline with sum readout to appreciate the contribution of message passing. We report the percentage of non-distinguished pairs in Figure 7b. Contrary to 3-WL, both CIN and the MLP baseline are able to distinguish many pairs across all families, with better performance attained for larger $k$. For $k = 6$, we observed CIN to disambiguate all pairs in all families (0.0% failure rate). Despite the strong results achieved by the baseline, we found CIN to always distinguish a larger number of non-isomorphic pairs for the same values of $k$, this confirming the importance of cellular message passing.

**RingTransfer** In order to empirically validate the ability of CIN to capture long-range node dependencies, we additionally design a third synthetic benchmark dubbed as 'RingTransfer'. Graphs in this dataset are chordless cycles (rings) of size $k$. In each graph we mark two special nodes as **target** and **source**, always placed at distance $\lfloor \frac{k}{2} \rfloor$. The task is for **target** to output the one-hot encoded label assigned to **source**. All other nodes in the ring are assigned a unitary constant feature vector. A model has to learn to transfer the information contained in **source** to the opposite side of the ring, where **target** resides. We initialise 1- and 2-dimensional cells with a null signal. In Figure 7a we show the performance of a 3-layer CIN as a function of the ring size $k$, along with that of GIN [74] baselines equipped with $\lfloor \frac{k}{2} \rfloor$ stacked layers. We observe that our model learns to solve the task with only 3 computational steps, independent of $k$. As for GIN, we observed degradation in the performance for $k \geq 24$, up to complete failure. We hypothesise this to be due to the difficulties of training such a deep GNN ($\geq 12$ layers). We further verify the (theoretically expected) failure of GIN (not included) when endowed with less than $\lfloor \frac{k}{2} \rfloor$ layers.

## 5.2 Real-World Graph Benchmarks

**TUD** We test our model on 8 TUDataset benchmarks [56] with small and medium sizes from biology (**PROTEINS** [9, 23]), chemistry (i.e. molecules – **MUTAG** [45, 61], **PTC**, **NCI1** and **NCI109** [70]) to social networks (**IMDB-B**, **IMDB-M**, **RDT-B**). We consider induced cycle of size up to $k = 6$ for our graph lifting procedure. We initialise node (and 0-cell) features as described in Xu et al. [74], and higher dimensional cells by averaging or summing the features of the included 0-cells. The training setting and evaluation procedure follow those in Xu et al. [74]. We report the results in Table 2. CIN compares more than favourably with the baselines, displaying strong empirical performance on all benchmarks. The mean accuracy of CIN ranks top on four out of eight datasets.

---

[3]Data available at: `http://users.cecs.anu.edu.au/~bdm/data/graphs.html`.

Table 2: TUDatasets. The first section of the table includes the accuracy of graph kernel methods, while the second includes GNNs. The top three are highlighted by **First**, **Second**, **Third**.

| Dataset | MUTAG | PTC | PROTEINS | NCI1 | NCI109 | IMDB-B | IMDB-M | RDT-B |
|---------|-------|-----|----------|------|--------|--------|--------|-------|
| RWK [29] | 79.2±2.1 | 55.9±0.3 | 59.6±0.1 | >3 days | N/A | N/A | N/A | N/A |
| GK ($k = 3$) [64] | 81.4±1.7 | 55.7±0.5 | 71.4±0.31 | 62.5±0.3 | 62.4±0.3 | N/A | N/A | N/A |
| PK [58] | 76.0±2.7 | 59.5±2.4 | 73.7±0.7 | 82.5±0.5 | N/A | N/A | N/A | N/A |
| WL kernel [65] | 90.4±5.7 | 59.9±4.3 | 75.0±3.1 | **86.0**±1.8 | N/A | 73.8±3.9 | 50.9±3.8 | 81.0±3.1 |
| DCNN [3] | N/A | N/A | 61.3±1.6 | 56.6±1.0 | N/A | 49.1±1.4 | 33.5±1.4 | N/A |
| DGCNN [76] | 85.8±1.8 | 58.6±2.5 | 75.5±0.9 | 74.4±0.5 | N/A | 70.0±0.9 | 47.8±0.9 | N/A |
| IGN [52] | 83.9±13.0 | 58.5±6.9 | **76.6**±5.5 | 74.3±2.7 | **72.8**±1.5 | 72.0±5.5 | 48.7±3.4 | N/A |
| GIN [74] | 89.4±5.6 | 64.6±7.0 | 76.2±2.8 | 82.7±1.7 | N/A | 75.1±5.1 | 52.3±2.8 | **92.4**±2.5 |
| PPGNs [53] | **90.6**±8.7 | 66.2±6.6 | **77.2**±4.7 | 83.2±1.1 | **82.2**±1.4 | 73.0±5.8 | 50.5±3.6 | N/A |
| Natural GN [21] | 89.4±1.6 | **66.8**±1.7 | 71.7±1.0 | 82.4±1.3 | N/A | 73.5±2.0 | 51.3±1.5 | N/A |
| GSN [10] | **92.2** ± 7.5 | **68.2** ± 7.2 | **76.6** ± 5.0 | **83.5** ± 2.0 | N/A | **77.8** ± 3.3 | **54.3** ± 3.3 | N/A |
| SIN [8] | N/A | N/A | 76.4 ± 3.3 | 82.7 ± 2.1 | N/A | **75.6** ± 3.2 | 52.4 ± 2.9 | **92.2** ± 1.0 |
| **CIN** (Ours) | **92.7** ± 6.1 | **68.2** ± 5.6 | **77.0** ± 4.3 | **83.6** ± 1.4 | **84.0** ± 1.6 | **75.6** ± 3.7 | **52.7** ± 3.1 | **92.4** ± 2.1 |

Table 3: ZINC (MAE), ZINC-FULL (MAE) and Mol-HIV (ROC-AUC).

| Method | ZINC ↓ | | ZINC-FULL ↓ | MOLHIV ↑ |
|--------|--------|--|-------------|----------|
| | No Edge Feat. | With Edge Feat. | All methods | All methods |
| GCN [47] | 0.469±0.002 | N/A | N/A | 76.06±0.97 |
| GAT [69] | 0.463±0.002 | N/A | N/A | N/A |
| GatedGCN [11] | 0.422±0.006 | 0.363±0.009 | N/A | N/A |
| GIN [74] | 0.408±0.008 | 0.252±0.014 | 0.088±0.002 | 77.07±1.49 |
| PNA [20] | 0.320±0.032 | 0.188±0.004 | N/A | 79.05±1.32 |
| DGN [6] | 0.219±0.010 | 0.168±0.003 | N/A | 79.70±0.97 |
| HIMP [27] | N/A | 0.151±0.006 | 0.036±0.002 | 78.80±0.82 |
| GSN [10] | 0.139±0.007 | 0.108±0.018 | N/A | 77.99±1.00 |
| **CIN-small** (Ours) | 0.139±0.008 | 0.094±0.004 | 0.044±0.003 | 80.55±1.04 |
| **CIN** (Ours) | **0.115±0.003** | **0.079±0.006** | **0.022±0.002** | **80.94±0.57** |

On the remaining datasets, CIN achieves the second place. We observe that the best results are on datasets from the biological and chemical domains, where rings play a relevant role.

**ZINC**    We study the effectiveness of cellular message passing on larger scale molecular benchmarks from the ZINC database [68]. **ZINC** (12k graphs) and **ZINC-FULL** (250k graphs) [24, 33, 43, 75] are two graph regression task datasets for drug constrained solubility prediction. In these experiments, we consider rings up to size $k = 18$. We follow the training and evaluation procedures in [24]. Our experiments encompass different scenarios, examine the impact of ablating edge features and of constraining the parameter budget of the architecture to 100k. All results are illustrated in Table 3 where we also include the results for **ZINC-FULL** obtained by the same exact architectures. Our model exhibits particularly strong performance on these benchmarks: it attains state-of-the-art results on both the two dataset variants, outperforming other models by a significant margin. CIN attains strong results even when constrained by the parameter budget. It still achieves state-of-the-art performance on **ZINC** and is on-par with the best unconstrained baseline under edge-feature ablation.

**Mol-HIV**    We additionally test our model on the molecular **ogbg-molhiv** dataset from the Open Graph Benchmark [40] (41k graphs). The task is to predict the capacity of compounds to inhibit HIV replication. Rings of size up to $k = 6$ are considered as 2-cells. We take the architecture in [27] as reference and replicate the same hyperparameter setting in our model, including the use of only 2 message passing layers. We report the mean of test ROC-AUC metrics at the epoch of best validation performance for 10 random weight initialisations. Similarly to ZINC, we experiment with a "small" model whose number of parameters is constrained in the order of 100k. Table 3 displays the results. CIN significantly outperforms other strong GNN baselines, even when constrained by the parameter budget. Consistently with [27], we observe that only two layers are sufficient when performing hierarchical message passing across meso-scale structures such as rings.

# 6 Related Work, Discussion and Conclusion

**Cell complex models**   Recent works have proposed the generalisation of GNNs to simplicial complexes [13, 25, 32, 35]. All these simplicial methods are subsumed by the model in Bodnar et al. [8], which CWNs in turn subsume. To the best of our knowledge, Hajij et al. [34] is the only other example of message passing on cell complexes, but this work does not study the expressive power of the proposed scheme, neither it experimentally validates its performance. In contrast, our work comprehensively characterises the expressiveness of cellular message passing, and introduces a theoretically grounded and empirically effective framework to apply it on graph structured data in a way to address several limitations of standard Graph Neural Networks.

**Molecular substructures**   A few other works have extended GNNs to account for molecular substructures. Junction Trees (JT), which conveniently represent singletons, bonds and rings as supernodes in a tree, have been used in molecular graph generation [43, 44]. JTs are also used in the recent work of Fey et al. [27], who employs them to design a hierarchical message passing scheme based on the tree structure. However, this hierarchy has a different configuration than the one cell complexes provide. Information about cycles is also used in GSNs [10] to augment the node features, but the model retains the usual message passing procedure of GNNs. These last two models are of particular relevance to the present work, since they utilise information about chemical rings. It is important to remark that CWNs compare favourably with both of them in all our benchmarks.

**Higher-order GNNs**   A related line of work has studied lifting graphs into $k$-dimensional tensor representations that can be processed by provably expressive $k$-GNNs [4, 52, 53]. With higher values of $k$, these models achieve higher-expressivity, but due to the computational complexity this incurs, values of $k \geq 3$ are of little use in practice. Therefore, unlike CWNs, these models cannot explicitly represent in practice chemical rings of common sizes (e.g. five or six). Furthermore, by being upper-bounded by 3-WL, the 2-GNN models cannot count the number of induced cycles of size greater than four (see Appendix A for details). In contrast, CWNs can easily count these important chemical substructures through the readout operation it performs on the 2-cells.

**Limitations**   The main limitations of the model are of computational nature. While the computational complexity of the message passing procedure and its preprocessing step is suitable for molecular and geometric graphs, the number of rings (and more generally simple cycles) in general graphs can be exponential in the number of nodes. In that case, one has to resort to smaller 2-cells like triangles, which can be found efficiently in general graphs. Moreover, one has to typically use weights specific for each dimension of the cell complex, increasing the number of parameters compared to GNNs. However, we have shown that our model can compensate this increase with a reduced number of layers and still achieve state-of-the-art results on some of the molecular benchmarks.

From a theoretical point of view, this work is concerned only with *regular* cell complexes. Adopting this restriction is useful from multiple perspectives: regular cell complexes are easier to analyse, their combinatorial structure completely describes their topology and convolutions can be defined on them through the Sheaf Laplacian (see Appendix D). Nonetheless, some of our theoretical results could be extended to non-regular complexes, which could be obtained by lifting transformations not studied in this work, such as attaching 2-cells to paths in the graph. We leave addressing non-regular complexes and their trade-offs to future developments of this work.

**Societal Impacts**   Most of our paper is theoretical in nature and we do not see immediate direct negative societal impacts. Within the scope of social network applications, we do not yet have sufficient evidence of performance improvement on related benchmarks to justify obvious adoption in such a domain. In contrast, the empirical performance on molecular benchmarks suggests it may have a positive impact on applications of immediate interest in pharmaceutics, such as drug discovery [30].

**Conclusion**   We have proposed a provably powerful message passing procedure on cell complexes motivated by a novel colour refinement algorithm to test their isomorphism. This allows us to consider flexible lifting operations on graphs to implement more expressive architectures which benefit from decoupling the computational and input graphs. Our methods show excellent performance on diverse synthetic and real-world molecular benchmarks.

## Funding and Acknowledgements

YW and GM acknowledge support from the ERC under the EU's Horizon 2020 programme (grant agreement n° 757983). MB is supported in part by ERC Consolidator grant n° 724228 (LEMAN). The authors declare no competing interests. We are also grateful to Ben Day, Gabriele Corso and Nikola Simidjievski for their helpful feedback. We would also like to thank Vijay P. Dwivedi and Chaitanya K. Joshi for clarifying certain aspects of their Benchmarking GNNs [24] work, and to Muhammet Balcilar for signalling a numerical precision issue in early SR graphs experiments.

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
