# A Proofs

## A.1 Cellular WL Results

In this section, we assume basic familiarity with the WL test and its higher-order variants. For an introduction to these topics, we refer the reader to the survey of Sato [62]. We begin by introducing a few useful concepts.

**Definition 20.** *A **cellular colouring** is a map $c$ that maps a cell complex $X$ and one of its cells $\sigma$ to a colour from a fixed colour palette. We denote this colour by $c_\sigma^X$.*

**Definition 21.** *Let $X, Y$ be two regular cell complexes and $c$ a cellular colouring. We say that $X, Y$ are **c-similar**, denoted by $c^X = c^Y$, if the number of cells in $X$ coloured with a given colour equals the number of cells in $Y$ with the same colour. Otherwise, we have $c^X \neq c^Y$.*

We emphasise that in this paper we are interested only in colourings $c$ with the property that any two isomorphic cell complexes are $c$-similar.

**Definition 22.** *A cellular colouring $c$ **refines** a cellular colouring $d$, denoted by $c \sqsubseteq d$, if for all cell complexes $X$ and $Y$ and all $\sigma \in P_X$ and $\tau \in P_Y$, $c_\sigma^X = c_\tau^Y$ implies $d_\sigma^X = d_\tau^Y$. Additionally, if $d \sqsubseteq c$, we say the two colourings are equivalent and we represent it by $c \equiv d$.*

We state the following result from Bodnar et al. [8] about simplicial colourings, which we translate here directly to cell complexes. The proof is however, identical, and we refer the reader to their work for that.

**Proposition 23.** *Let $X, Y$ be any regular cellular complexes with $A \subseteq P_X$ and $B \subseteq P_Y$. Consider two cellular colourings $c, d$ such that $c \sqsubseteq d$. If $\{\!\{d_\sigma^X \mid \sigma \in A\}\!\} \neq \{\!\{d_\tau^Y \mid \tau \in B\}\!\}$, then $\{\!\{c_\sigma^X \mid \sigma \in A\}\!\} \neq \{\!\{c_\tau^Y \mid \tau \in B\}\!\}$.*

**Corollary 24.** *Consider two cellular colourings $c, d$ such that $c \sqsubseteq d$. For all cell complexes $X$ and $Y$, if $d^X \neq d^Y$, then $c^X \neq c^Y$.*

This last result implies that if $c$ refines $d$, then $c$ can distinguish all the non-isomorphic cell complexes that $d$ can distinguish. We say that the colouring $c$ is at least as powerful as the colouring $d$.

In contrast to simplicial complexes, cell complexes have a more flexible structure. The main complication compared to the proofs in Bodnar et al. [8] is that cells can have a variable number of lower-dimensional cells on their boundary. It is therefore useful in many proofs, to separate the cells into buckets containing cells with the same boundary size. The following result helps us do that.

**Proposition 25.** *Let $c^t$ be the CWL colouring at iteration $t$. For all cells $\sigma, \tau$ in any cell complexes $X$ and $Y$, if $|\mathcal{B}(\sigma)| \neq |\mathcal{B}(\tau)|$, then for any $t > 0$ we have $c_\sigma^t \neq c_\tau^t$.*

*Proof.* If $\sigma$ and $\tau$ have boundaries of different sizes, then $c_{\mathcal{B}}^1(\sigma) \neq c_{\mathcal{B}}^1(\tau)$, which immediately implies $c_\sigma^t \neq c_\tau^t$ for all $t > 0$. $\square$

Next, we show that one can drop the co-boundary adjacencies without sacrificing expressive power.

**Lemma 26.** *CWL with $\mathrm{HASH}\big(c_\sigma^t, c_{\mathcal{B}}^t(\sigma), c_\downarrow^t(\sigma), c_\uparrow^t(\sigma)\big)$ is as powerful as CWL with the generalised update rule $\mathrm{HASH}\big(c_\sigma^t, c_{\mathcal{B}}^t(\sigma), c_{\mathcal{C}}^t(\sigma), c_\downarrow^t(\sigma), c_\uparrow^t(\sigma)\big)$.*

*Proof.* Let $a^t$ denote the colouring produced by CWL using the general version and $b^t$ the colouring produced using the restricted version at iteration $t$. It can be verified that $a^t \sqsubseteq b^t$ because it considers the additional $c_{\mathcal{B}}^t(\sigma)$ colours in the refinement rule. We now prove $b^{t+1} \sqsubseteq a^t$ by induction. Note that to take advantage of Proposition 25, we shift the time-step by one (i.e. we use $b^{t+1}$ as opposed to $b^t$).

The base case holds since $a^0$ assigns the same colour to all the cells. Suppose $b_\sigma^{t+2} = b_\tau^{t+2}$ for any two cells $\sigma$ and $\tau$ from any cell complexes $X$ and $Y$, respectively. Then we know that $b_\sigma^{t+1} = b_\tau^{t+1}, b_{\mathcal{B}}^{t+1}(\sigma) = b_{\mathcal{B}}^{t+1}(\tau), b_\uparrow^{t+1}(\sigma) = b_\uparrow^{t+1}(\tau)$ and $b_\downarrow^{t+1}(\sigma) = b_\downarrow^{t+1}(\tau)$. The goal is to show that this also implies that $b_{\mathcal{C}}^{t+1}(\sigma) = b_{\mathcal{C}}^{t+1}(\tau)$.

Given $b_\uparrow^{t+1}(\sigma) = b_\uparrow^{t+1}(\tau)$, by definition

$$\{\!\{b_{\delta_\sigma}^{t+1} \mid (\cdot, b_{\delta_\sigma}^{t+1}) \in b_\uparrow^{t+1}(\sigma)\}\!\} = \{\!\{b_{\delta_\tau}^{t+1} \mid (\cdot, b_{\delta_\tau}^{t+1}) \in b_\uparrow^{t+1}(\tau)\}\!\}.$$

By Proposition 25, cells with different boundary sizes have different colours. Therefore, we can partition these two multi-sets by the size of the cell boundaries, while preserving the equality between these sub-multisets. Therefore, for each $n \in \mathbb{N}$:

$$\{\!\{ b_{\delta_\sigma}^{t+1} \mid (\cdot, b_{\delta_\sigma}^{t+1}) \in b_\uparrow^{t+1}(\sigma) \text{ and } |\mathcal{B}(\delta_\sigma)| = n \}\!\} = \{\!\{ b_{\delta_\tau}^{t+1} \mid (\cdot, b_{\delta_\tau}^{t+1}) \in b_\uparrow^{t+1}(\tau) \text{ and } |\mathcal{B}(\delta_\tau)| = n \}\!\}.$$

Let $\gamma$ be an arbitrary cell. Then for each cell $\delta_\gamma \in \mathcal{C}(\gamma)$, $\gamma$ exchanges messages with all the other boundary cells of $\delta_\gamma$. Therefore, the colour of each $\delta_\gamma$ with $|\mathcal{B}(\delta_\gamma)| = n$ shows up with a multiplicity of $n-1$ in the tuples of $b_\uparrow^{t+1}(\gamma)$. Eliminating $n-2$ of these repeated colours for all $\delta_\sigma$ and $\delta_\tau$:

$$\{\!\{ b_{\delta_\sigma}^{t+1} \mid \delta_\sigma \in \mathcal{C}(\sigma) \text{ and } |\mathcal{B}(\delta_\sigma)| = n \}\!\} = \{\!\{ b_{\delta_\tau}^{t+1} \mid \delta_\tau \in \mathcal{C}(\tau) \text{ and } |\mathcal{B}(\delta_\tau)| = n \}\!\}.$$

Merging these in a single multi-set gives the colours of the co-boundary cells:

$$b_\mathcal{C}^{t+1}(\sigma) = \{\!\{ b_{\delta_\sigma}^{t+1} \mid \delta_\sigma \in \mathcal{C}(\sigma) \}\!\} = \{\!\{ b_{\delta_\tau}^{t+1} \mid \delta_\tau \in \mathcal{C}(\tau) \}\!\} = b_\mathcal{C}^{t+1}(\tau).$$

By the induction hypothesis, $a_\sigma^t = a_\tau^t, a_\mathcal{B}^t(\sigma) = a_\mathcal{B}^t(\tau), a_\mathcal{C}^t(\sigma) = a_\mathcal{C}^t(\tau), a_\uparrow^t(\sigma) = a_\uparrow^t(\tau)$ and $a_\downarrow^t(\sigma) = a_\downarrow^t(\tau)$. This implies $a_\sigma^{t+1} = a_\tau^{t+1}$. $\qquad\square$

The following theorem shows that we can further prune the CWL update rule by removing the colours associated with the lower adjacencies. The structure of the proof is similar to the one in Bodnar et al. [8], with the main difference being in the proof of Proposition 27.

***Proof of Theorem* 7** Let $b^t$ denote the colouring of CWL using $\text{HASH}\big(b_\sigma^t, b_\mathcal{B}^t(\sigma), b_\uparrow^t(\sigma)\big)$ and $a^t$ the colouring of CWL using the rule $\text{HASH}\big(a_\sigma^t, a_\mathcal{B}^t(\sigma), a_\downarrow^t(\sigma), a_\uparrow^t(\sigma)\big)$ from Lemma 26. Trivially $a^t \sqsubseteq b^t$ because of the additional argument $c_\downarrow^t(\sigma)$ in the update rule. We prove $b^{2t+1} \sqsubseteq a^t$ by induction. As before, the addition by one in $2t+1$ is to allow us to apply Proposition 25 in the induction step. The multiplication by 2 is due to the fact that the information transmitted through the lower adjacencies in one step is propagated in two steps through the boundary adjacencies.

As before, the base case trivially holds since $a^0$ assigns the same colour to all cells. Suppose $b_\sigma^{2t+3} = b_\tau^{2t+3}$. By unrolling the hash function two steps in time, we obtain $b_\sigma^{2t+1} = b_\tau^{2t+1}$, $b_\mathcal{B}^{2t+1}(\sigma) = b_\mathcal{B}^{2t+1}(\tau)$, and $b_\uparrow^{2t+1}(\sigma) = b_\uparrow^{2t+1}(\tau)$. We need to prove that $b_\downarrow^{2t+1}(\sigma) = b_\downarrow^{2t+1}(\tau)$ also holds. For the sake of contradiction, assume $b_\downarrow^{2t+1}(\sigma) \neq b_\downarrow^{2t+1}(\tau)$. Then there exists a pair of colours $(\mathbb{C}_0, \mathbb{C}_1)$ that shows up (without loss of generality) more times in $b_\downarrow^{2t+1}(\sigma)$ than in $b_\downarrow^{2t+1}(\tau)$. For simplicity, we also assume $b_\sigma^{2t+1} \neq \mathbb{C}_0 \neq b_\tau^{2t+1}$ as this special case can be easily treated separately.

For all cell complexes $X$ and all cells $\delta$ in $P_X$, consider the collection of multi-sets $A_X$ indexed by $\delta$:

$$A_X(\delta) = \{\!\{ (b_\psi^{2t+1} = \mathbb{C}_0, b_\delta^{2t+1} = \mathbb{C}_1) \mid \psi \in \mathcal{C}(\delta) \}\!\}.$$

We are interested in the size of these multi-sets for some specific cells $\delta$. To that end, for each cell $\gamma \in P_X$, we define the multi-set:

$$C_X(\gamma) = \{\!\{ |A_X(\delta)| \mid \delta \in \mathcal{B}(\gamma) \}\!\}.$$

We know that $C_X(\sigma) \neq C_Y(\tau)$ since the sum of the elements of $C_X(\sigma)$, which gives the number of tuples $(\mathbb{C}_0, \mathbb{C}_1)$ in $b_\downarrow^{2t+1}(\sigma)$, is greater than the sum of the elements of $C_Y(\tau)$, which gives the number of tuples $(\mathbb{C}_0, \mathbb{C}_1)$ in $b_\downarrow^{2t+1}(\tau)$. We prove this contradicts our hypothesis that $b_\sigma^{2t+3} = b_\tau^{2t+3}$.

**Proposition 27.** *For all regular cell complexes $X, Y$ and all $\sigma \in P_X, \tau \in P_Y$, if $C_X(\sigma) \neq C_Y(\tau)$, then $b_\sigma^{2t+3} \neq b_\tau^{2t+3}$.*

*Proof.* Given a cell complex $X$ and a cell $\delta \in P_X$, consider the cellular colouring $c_\delta^X = |A_X(\delta)|$. The idea of the proof is to show that $b^{2t+2} \sqsubseteq c$, which allows us to use Proposition 23 for the multi-sets $C_X(\sigma)$ and $C_Y(\tau)$.

Let $\delta_1, \delta_2$ be two arbitrary cells from any regular cell complexes $X, Y$ such that $c_{\delta_1}^X \neq c_{\delta_2}^Y$. Assume without loss of generality that $|A_X(\delta_1)| > |A_Y(\delta_2)|$. Two cases can be distinguished for this inequality. In the first case, $b_{\delta_2}^{2t+1} \neq \mathbb{C}_1$, which implies $|A_X(\delta_1)| > |A_Y(\delta_2)| = 0$ and, therefore, $b_{\delta_1}^{2t+1} = \mathbb{C}_1$. Then $b_{\delta_1}^{2t+2} \neq b_{\delta_2}^{2t+2}$.

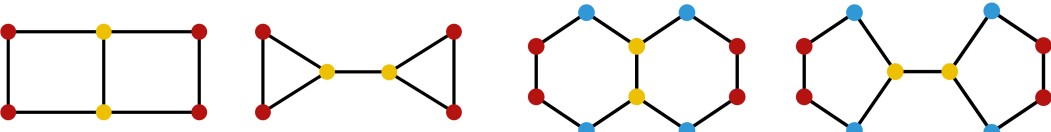

Figure 8: (Left) A pair of non-isomorphic graphs indistinguishable by WL, but distinguishable by CWL with a clique complex, ring or cycle-based lifting. (Right) A pair of non-isomorphic molecular graphs (Decalin and Bicyclopentyl) indistinguishable by WL but distinguishable by CWL with a ring-based or cycle-based lifting. The node colours show the stable colouring reached by WL.

In the second case, $b_{\delta_2}^{2t+1} = \mathbb{C}_1$, which implies $|A_X(\delta_1)| > |A_Y(\delta_2)| \geq 0$ and $b_{\delta_1}^{2t+1} = \mathbb{C}_1$. Then, the difference in the size of the multi-sets is made by the number of times $\mathbb{C}_0$ shows up in $A_X(\delta_1)$ and $A_Y(\delta_2)$, respectively. By Proposition 25, all $k$-cells $\gamma$ with $k > 0$ and $b_\gamma^{2t+1} = \mathbb{C}_0$ must have a fixed boundary size $|\mathcal{B}(\gamma)| = n$. Because each cell $\delta \in \mathcal{B}(\gamma)$ is upper adjacent with every other cell in $\mathcal{B}(\gamma)$, $b_\gamma^{2t+1}$ appears $n - 1$ times in the tuples inside $b_\uparrow^{2t+1}(\delta)$. Additionally, note that since the cell complex is regular, self-loops are not allowed and, therefore, $n > 1$.

Applying this to $\delta_1$ and $\delta_2$, $\mathbb{C}_0$ shows up $|A_X(\delta_1)| \times (n-1)$ times in $b_\uparrow^{2t+1}(\delta_1)$ and $|A_Y(\delta_2)| \times (n-1)$ times in $b_\uparrow^{2t+1}(\delta_2)$. Therefore, $b_\uparrow^{2t+1}(\delta_1) \neq b_\uparrow^{2t+1}(\delta_2)$ and, similarly to the first case, $b_{\delta_1}^{2t+2} \neq b_{\delta_2}^{2t+2}$. The results obtained for the two cases prove $b^{2t+2} \sqsubseteq c$.

Applying Proposition 23 for the multi-sets $C_X(\sigma)$ and $C_Y(\tau)$, we obtain two non-equal multi-sets:

$$b_\mathcal{B}^{2t+2}(\sigma) = \{\!\{b_{\delta_1}^{2t+2} \mid \delta_1 \in \mathcal{B}(\sigma)\}\!\} \neq \{\!\{b_{\delta_2}^{2t+2} \mid \delta_2 \in \mathcal{B}(\tau)\}\!\} = b_\mathcal{B}^{2t+2}(\tau)$$

Since these two multi-sets are used in the colour updating rule, $b_\sigma^{2t+3} \neq b_\tau^{2t+3}$. $\square$

Therefore, $b_\downarrow^{2t+1}(\sigma) = b_\downarrow^{2t+1}(\tau)$. Finally, applying the induction hypothesis, we have that $a_\sigma^t = a_\tau^t, a_\mathcal{B}^t(\sigma) = a_\mathcal{B}^t(\tau), a_\uparrow^t(\sigma) = a_\uparrow^t(\tau)$ and $a_\downarrow^t(\sigma) = a_\downarrow^t(\tau)$. Then $a_\sigma^{t+1} = a_\tau^{t+1}$. $\square$

***Proof of Theorem 13.*** Consider the map $f : \mathcal{G} \to \mathcal{X}$, a skeleton-preserving lifting transformation from the space of graphs $\mathcal{G}$, to the space of regular cell complexes $\mathcal{X}$. Let $g_G : V_G \to P_{f(G)^{(0)}}$ be the graph isomorphism associated to $f$ between the vertices of $G$ and the 0-cells of $f(G)$ for all $G \in \mathcal{G}$. Let $c^{G,t}$ be the WL colouring of graph $G$ at iteration $t$ and $a^{f(G),t}$ the colouring of $f(G)^{(1)}$ induced by the isomorphism $g_G$ (i.e $a_{g(v)}^{f(G)^{(1)},t} := c_v^{G,t}$) at the same time step $t$.

Because $f(G)^{(1)}$ and $G$ are isomorphic as graphs and WL is invariant under isomorphism, $a^{f(G)^{(1)},t} = c^{G,t}$. It follows that for all graphs $G_1, G_2 \in \mathcal{G}$, if $c^{G_1,t} \neq c^{G_2,t}$ then $a^{f(G_1)^{(1)},t} \neq^{f(G_2)^{(1)},t}$. Let $b^t$ be the CWL colouring of the 0-cells at iteration $t$. The goal is to show that for all regular cell complexes $X, Y \in f(\mathcal{G})$, $b^t \sqsubseteq a^t$. By transitivity and combined with Corollary 24, it follows that if $c^{G_1,t} \neq c^{G_2,t}$, then $b^{f(G_1),t} \neq b^{f(G_2),t}$.

The base case trivially holds. Let $\sigma, \tau$ be two 0-cells in $X \in f(\mathcal{G})$ and $Y \in f(\mathcal{G})$, respectively such that $b^{t+1}(\sigma) = b^{t+1}(\tau)$. Since 0-cells have only upper adjacencies, the equality implies that $b_\sigma^t = b_\tau^t$ and $b_\uparrow^t(\sigma) = b_\uparrow^t(\tau)$. The latter multi-set equality further implies

$$\{\!\{b_{\delta_\sigma}^t \mid (b_{\delta_\sigma}^t, \cdot) \in b_\uparrow(\sigma)\}\!\} = \{\!\{b_{\delta_\tau}^t \mid (b_{\delta_\tau}^t, \cdot) \in b_\uparrow(\tau)\}\!\}.$$

Equivalently, for 0-cells of a cell complex whose 1-skeleton is a graph (i.e. not a multi-graph), this can be rewritten as

$$\{\!\{b_{\delta_\sigma}^t \mid \delta_\sigma \in \mathcal{N}_\uparrow(\sigma)\}\!\} = \{\!\{b_{\delta_\tau}^t \mid \delta_\tau \in \mathcal{N}_\uparrow(\tau)\}\!\}.$$

By the induction hypothesis we have $a_\sigma^t = a_\tau^t$ and

$$\{\!\{a_{\delta_\sigma}^t \mid \delta_\sigma \in \mathcal{N}_\uparrow(\sigma)\}\!\} = \{\!\{a_{\delta_\tau}^t \mid \delta_\tau \in \mathcal{N}_\uparrow(\tau)\}\!\}.$$

These equalities imply $a^{t+1}(\sigma) = a^{t+1}(\tau)$. $\square$

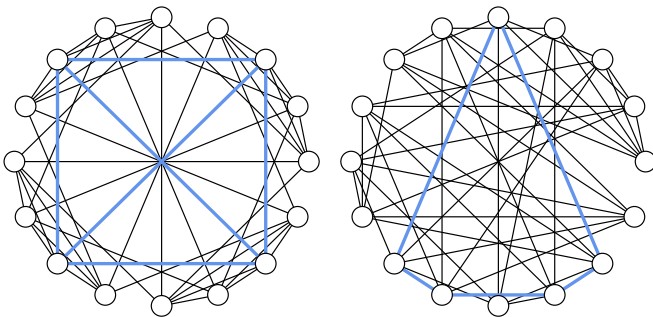

Figure 9: The two SR graphs in family $SR(16, 6, 2, 2)$: Rook's $4\times4$ (left) and Shrikhande (right). The 3-WL test is not able to deem them as non-isomorphic. Contrary to the Shrikhande graph, Rook's graph possesses 4-cliques. The Shrikhande graph, however, features 5-rings, not present in Rook's. Instances of these substructures are marked in blue. With appropriate lifting procedures, CWL can distinguish between them.

***Proof of Corollary 15.*** Due to Theorem 13, it is sufficient to find some examples of non-isomorphic graph pairs that WL cannot distinguish, but CWL can with the given lifting transformations. Figure 8 includes such examples. Based on Proposition 25, CWL can distinguish these graphs since it can count the number of substructures (e.g. triangles, rings, cycles) that the lifting is based on. □

The next proposition shows that CWL can identify cells that are $n$-simplices.

**Proposition 28** (Simplex Identification). *Let $X, Y$ be regular cell complexes and $\sigma \in P_X, \tau \in P_Y$ two cells. Denote by $c^t$ the CWL colouring at iteration $t$. Suppose $\sigma$ is an $n$-simplex and $\tau$ is not. Then $c^t(\sigma) \neq c^t(\tau)$ for all $t \geq n + 1$.*

*Proof.* The base case holds since $c^1_\sigma \neq c^1_\tau$ if $\sigma$ is a vertex and $\tau$ is a cell of another dimension. This is because $\sigma$ has no boundary adjacencies, while $\tau$ does.

Suppose the statement holds for $n$-simplices. Then, an $(n+1)$-simplex can be identified by having $n + 2$ $n$-simplices on its boundary. By Proposition 25, the colour of $\sigma$ encodes the boundary size. Furthermore, by the induction hypothesis $c^{n+1}_{\mathcal{B}}(\sigma)$ encodes the fact that the boundary cells are $n$-simplices. □

***Proof of Theorem 16.*** The sub-results of the theorem can be proven by finding pairs of graphs from the same family of Strongly Regular Graphs that can be distinguished by CWL with the corresponding lifting transformations. Graphs in this family are provably indistinguishable by the higher-order 3-WL test [8].

**Ring-based lifting**   We can show that there is a pair of SR graphs in the same family with a different number of induced cycles of a certain size. We include such an example in Figure 9. The two graphs differ in the number of 4-, 5-, 6- and 8-rings (see Table 4), which indirectly proves 3-WL cannot count induced cycles of these sizes. It is also natural to conjecture that 3-WL cannot count induced cycles of size strictly larger than 3. In contrast, CWL(4-IC) is sufficient to distinguish these two graphs.

**Clique complex lifting**   We can leverage on the same example: the graph on the right does not possess 4-cliques, contrary to the graph on the left (one such example is marked in blue). This proves that 3-WL cannot count cliques of size 4. As shown by Bodnar et al. [8], this result immediately implies that SWL (and consequently CWL) with a clique complex lifting is not less powerful than 3-WL.

**Cycle-based lifting**   To prove the result for this lifting transformation we leverage on a result by Arvind et al. [2], who show that 2-Folklore WL (which is equivalent to 3-WL [62]) cannot count subgraph cycles of size strictly larger than 7. Table 4 illustrates this for the same example as above.

Table 4: Number of cycles and induced cycles (rings) on the SR graphs in family SR$(16, 6, 2, 2)$.

| Graph ↓ / Size → | 3 (Tri.) | 4 | 5 | 6 | 7 | 8 |
|---|---|---|---|---|---|---|
| Rook's 4×4 (cycles) | 32 | 60 | 288 | 1,248 | 4,032 | 11,952 |
| Shrikhande (cycles) | 32 | 60 | 288 | 1,248 | 4,032 | 11,688 |
| Rook's 4×4 (rings) | 32 | 36 | 0 | 96 | 0 | 72 |
| Shrikhande (rings) | 32 | 12 | 96 | 64 | 0 | 36 |

Since CWL can count the number of 8-cycles when the lifting transformation $k$-C with $k \geq 8$ is used (see Proposition 25), this proves the result.

$\square$

We note that while the proof above is purely based on substructure counts, the superior expressive power of CWL is very likely not limited to counting the substructures involved in the lifting transformation. We have seen evidence in favour of this claim in the SR experiment in Section 5, where message passing layers reduced the failure rate.

Next, we prove a statement comparing Simplicial WL and CWL. This will later be used to show that CWNs are strictly more powerful than MPSNs when a lifting transformation based on the clique complex and rings is used.

**Definition 29.** *A subset $L$ of a cell complex $X$ is called a **subcomplex** if it is a union of cells of $X$ containing the closures of these cells.*

**Theorem 30.** *Let $f : \mathcal{G} \to \mathcal{X}$ be a skeleton-preserving transformation such that for any graph $G$, the clique complex of $G$ is a subcomplex of $f(G)$. Then CWL($f$) is at least as powerful as SWL using the clique complex lifting at distinguishing non-isomorphic graphs.*

*Proof.* Let $c^t$ be the simplicial colouring performed by SWL. We can extend it into a cellular colouring $a^t$ defined as follows:

$$a_\sigma^{X,t} := \begin{cases} c_\sigma^{L,t} & \text{if } X_\sigma \subseteq L \\ \bigcirc & \text{otherwise} \end{cases}$$

where $L$ is the maximal simplicial complex that is a subcomplex of $X$ and $\bigcirc$ is a special colour assigned to the cells that are not simplices. Let $h : \mathcal{G} \to \mathcal{X}$ be the clique-complex lifting map. Then, it is easy to see that for all graphs $G_1, G_2 \in \mathcal{G}$, if $c^{h(G_1),t} \neq c^{h(G_2),t}$, then $a^{f(G_1),t} \neq a^{f(G_2),t}$. Let $b^t$ be the CWL colouring map at iteration $t$. We aim to show that $b^{t+n+1} \sqsubseteq a^t$ by using Proposition 28. Then, by transitivity and using Corollary 23, if $c^{h(G_1),t} \neq c^{h(G_2),t}$, then $b^{f(G_1),t+n+1} \neq b^{f(G_2),t+n+1}$.

Let $n$ be the maximum dimension of the cells used by the lifting transformation $f$. As usual, the base case holds at initialisation since $a^0$ assigns the same colour to all the cells. Let $\sigma, \tau$ be two cells from the regular cell complexes $X, Y \in f(\mathcal{G})$. When $\sigma$ and $\tau$ are not simplices, then $a_\sigma^t = a_\tau^t = \bigcirc$. Suppose $\sigma$ and $\tau$ are both simplices and $b_\sigma^{t+n+2} = b_\tau^{t+n+2}$. Then we know that $b_\sigma^{t+n+1} = b_\tau^{t+n+1}, b_\mathcal{B}^{t+n+1}(\sigma) = b_\mathcal{B}^{t+n+1}(\tau)$ and $b_\uparrow^{t+n+1}(\sigma) = b_\uparrow^{t+n+1}(\tau)$. Since $\sigma$ and $\tau$ are simplices, their boundary cells are also lower-dimensional simplices, so by induction hypothesis, $a_\mathcal{B}^t(\sigma) = a_\mathcal{B}^t(\tau)$.

Let us consider the equality between the colours involving the upper adjacent cells. By expanding the definition we have:

$$\{\!\{(b_{\delta_1}^{t+n+1}, b_{\delta_2}^{t+n+1}) \mid \delta_1 \in \mathcal{N}_\uparrow(\sigma), \delta_2 \in \mathcal{C}(\sigma, \delta_1)\}\!\}$$
$$= \{\!\{(b_{\delta_1}^{t+n+1}, b_{\delta_2}^{t+n+1}) \mid \delta_1 \in \mathcal{N}_\uparrow(\tau), \delta_2 \in \mathcal{C}(\tau, \delta_1)\}\!\}.$$

Generally, not all of these adjacencies involve simplices. For instance, a 2-simplex could incident to a general 3-cell. However, by Proposition 28 this equality must still hold if we restrict the multi-sets to the colour of those cells that are simplices:

$$\{\!\{(b_{\delta_1}^{t+n+1}, b_{\delta_2}^{t+n+1}) \mid \delta_1 \in \mathcal{N}_\uparrow(\sigma), \delta_2 \in \mathcal{C}(\sigma, \delta_1), \text{ and } \delta_1, \delta_2 \text{ are simplices}\}\!\}$$
$$= \{\!\{(b_{\delta_1}^{t+n+1}, b_{\delta_2}^{t+n+1}) \mid \delta_1 \in \mathcal{N}_\uparrow(\tau), \delta_2 \in \mathcal{C}(\tau, \delta_1), \text{ and } \delta_1, \delta_2 \text{ are simplices}\}\!\}.$$

These multi-sets, give exactly the upper adjacencies used by SWL for computing its colouring map $c^t$. Therefore, by the induction hypothesis, $a_\uparrow^t(\sigma) = a_\uparrow^t(\tau)$. Finally, this proves $a_\sigma^{t+1} = a_\tau^{t+1}$. □

**Corollary 31.** *CWL($k_1$-CL $\cup$ $k_2$-IC) and CWL($k_1$-CL $\cup$ $k_2$-C) are strictly more powerful than SWL($k_1$-CL) for all $k_2 \geq 5$.*

*Proof.* The second pair of graphs from Figure 8 cannot be distinguished by SWL($k_1$-CL) because it has no cliques greater than two, but it can be distinguished by CWL with the liftings above because of the different number of (induced) cycles. □

### A.2   CW Network Proof

***Proof of Theorem 18.*** Let $c^t$ denote the colouring of CWL at iteration $t$ and $h^t$ the colouring (i.e. features) produced by a CW-Network as described in Section 4. Without loss of generality (Theorem 7), we use only boundary and upper adjacencies for both methods.

To show CWNs are at most as powerful as CWL, we must show $c^t \sqsubseteq h^t$. Again, we show this by induction. For a CWN with $L$ layers we assume $h^t = h^L$ for all $t > L$. Let $\sigma, \tau$ be two cells with $c_\sigma^{t+1} = c_\tau^{t+1}$. Then, $c_\sigma^t = c_\tau^t$, $c_\mathcal{B}^t(\sigma) = c_\mathcal{B}^t(\tau)$ and $c_\uparrow^t(\sigma) = c_\uparrow^t(\tau)$. By the induction hypothesis, $h_\sigma^t = h_\tau^t$, $h_\mathcal{B}^t(\sigma) = h_\mathcal{B}^t(\tau)$ and $h_\uparrow^t(\sigma) = h_\uparrow^t(\tau)$.

If $t + 1 > L$, then $h_\sigma^{t+1} = h_\sigma^t = h_\tau^t = h_\tau^{t+1}$. Otherwise, $h^{t+1}$ is given by Equation 1 involving the update function $U$, the aggregate function AGG and the message functions $M_\mathcal{B}, M_\uparrow$. Given that the inputs passed to these functions are equal for $\sigma$ and $\tau$, $h_\sigma^{t+1} = h_\tau^{t+1}$.

We now prove that CWNs can be as powerful as CWL. Suppose the aggregation from Equation 1 is injective and the model is equipped with a number of layers $L$ sufficient to guarantee the convergence of the colouring. Then, we show that $h^t \sqsubseteq c^t$. Let $\sigma, \tau$ be two cells with $h_\sigma^{t+1} = h_\tau^{t+1}$. Then, since the local aggregation is injective $h_\sigma^t = h_\tau^t$, $h_\mathcal{B}^t(\sigma) = h_\mathcal{B}^t(\tau)$ and $h_\uparrow^t(\sigma) = h_\uparrow^t(\tau)$. By the induction hypothesis, $c_\sigma^t = c_\tau^t$, $c_\mathcal{B}^t(\sigma) = c_\mathcal{B}^t(\tau)$ and $c_\uparrow^t(\sigma) = c_\uparrow^t(\tau)$. Finally, $c_\sigma^{t+1} = c_\tau^{t+1}$. □

The consequence of this result is that CWNs inherit all the properties of CWL. We summarise these in the following Corollary.

**Corollary 32.** *CWNs have the following properties:*

1. *They are at least as powerful as the WL test when using skeleton-preserving lifting transformations.*

2. *They are strictly more powerful than the WL test when using the lifting maps from Corollary 15.*

3. *They are not less powerful than 3-WL when using the lifting transformations from Theorem 16.*

4. *They are at least as powerful as MPSNs using the clique complex lifting [8] when using a lifting transformation whose output complexes have the clique complex as a subcomplex.*

5. *They are strictly more powerful than MPSNs when using a transformation attaching cells to cliques and rings/cycles. In particular, CWNs using rings are strictly more powerful than MPSNs using a lifting based on triangles (i.e. 2-simplices), since triangles are rings of size 3.*

The latter point regarding triangles is important because Bodnar et al. [8] do not use simplices of dimension higher than two in practice.

### A.3   Equations for Other Adjacencies

For completeness, we include in this section the equations for the co-boundary and lower adjacent messages.

$$m_\mathcal{C}^{t+1}(\sigma) = \text{AGG}_{\tau \in \mathcal{C}(\sigma)}\Big( M_\mathcal{C}\big(h_\sigma^t, h_\tau^t\big)\Big), \quad m_\downarrow^{t+1}(\sigma) = \text{AGG}_{\tau \in \mathcal{N}_\downarrow(\sigma), \delta \in \mathcal{B}(\sigma,\tau)}\Big( M_\downarrow\big(h_\sigma^t, h_\tau^t, h_\delta^t\big)\Big).$$

Together with the adjacencies described in the main text, the update rule takes the form

$$h_\sigma^{t+1} = U\Big( h_\sigma^t, m_\mathcal{B}^t(\sigma), m_\mathcal{C}^t(\sigma), m_\downarrow^{t+1}(\sigma), m_\uparrow^{t+1}(\sigma)\Big).$$

As mentioned before, even though these adjacencies are redundant from a colour refinement perspective when the others are used, they might still be employed in other combinations that preserve the expressive power of the test. Additionally, for certain applications, they might still encode important inductive biases.

## B Computational Analysis

Let $X$ be a $d$-dimensional regular cell complex. For an arbitrary $p$-cell $\sigma$ with boundary size $k$, the number of $\uparrow$-messages between the $(p-1)$-cells on its boundary is $2 * \binom{k}{2}$ and the number of $\mathcal{B}$-messages it receives is $k$. Let $B_p$ be the maximum boundary size of a $p$-cell in $X$ and $S_p$ the number of $p$-cells. The computational complexity of our message passing scheme is thus $\mathcal{O}\big(\sum_{p=1}^{d} B_p S_p + 2 * \binom{B_p}{2} S_p\big)$. For instance, consider the skeleton-preserving lifting based on induced cycles. There, the dimension of the complex is $d = 2$ and we have $B_0 = 0$, $B_1 = 2$, and $B_2$ equals the size of the maximum induced cycle considered. For all practical purposes, we can consider $d$ and $B_p$ as fixed constants. Then the complexity can be rewritten as $\Theta\big(\sum_{p=1}^{d} S_p\big)$. This is optimal because the complexity is linear in the size of the cell complex and a linear time is required to read the cell complex.

Table 5: Wall-clock training and evaluation times on ZINC; mean, std over 10 runs (seconds).

| Model | Training (Epoch) | Eval (Train) | Eval (Val) | Eval (Test) |
|---|---|---|---|---|
| GIN | $4.582 \pm 0.012$ | $3.138 \pm 0.071$ | $0.310 \pm 0.002$ | $0.309 \pm 0.001$ |
| GIN-small | $3.737 \pm 0.012$ | $3.070 \pm 0.058$ | $0.304 \pm 0.002$ | $0.303 \pm 0.003$ |
| CIN | $10.828 \pm 0.059$ | $4.679 \pm 0.051$ | $0.470 \pm 0.002$ | $0.471 \pm 0.003$ |
| CIN-small | $7.082 \pm 0.041$ | $3.682 \pm 0.056$ | $0.365 \pm 0.002$ | $0.373 \pm 0.030$ |

Table 6: Wall-clock training and evaluation times on ZINC-FULL; mean, std over 10 runs (seconds).

| Model | Training (Epoch) | Eval (Train) | Eval (Val) | Eval (Test) |
|---|---|---|---|---|
| GIN | $106.268 \pm 1.991$ | $73.051 \pm 1.742$ | $7.874 \pm 0.174$ | $1.618 \pm 0.039$ |
| GIN-small | $87.581 \pm 2.343$ | $71.160 \pm 1.865$ | $7.714 \pm 0.206$ | $1.583 \pm 0.037$ |
| CIN | $249.334 \pm 17.927$ | $107.510 \pm 1.637$ | $11.759 \pm 0.642$ | $2.398 \pm 0.028$ |
| CIN-small | $163.282 \pm 8.016$ | $85.342 \pm 2.637$ | $9.251 \pm 0.431$ | $1.876 \pm 0.044$ |

In practice, we observed the empirical training runtimes to be contained, even on the largest benchmarks. We performed timing analyses on ZINC and ZINC-FULL, measuring the time required to complete one training epoch and a full performance evaluation on train, validation and test sets. We report the runtimes in Tables 5 and 6 the runtimes measured for our best performing CIN and CIN-small models and by GIN baselines with, approximately, the same number of parameters. We observe that the evaluation runtimes are relatively comparable to those of GIN models and that the difference decreases significantly at inference time (i.e. no backprop). The training runtimes are significantly reduced on CIN-small architectures, which always perform on-par or even better than state-of-the-art baselines, regardless of the imposed parameter budget (see Table 3). These experiments where run over an NVIDIA® Tesla V100 GPU device on an Amazon Web Services (AWS) Elastic Cloud (EC) 2 `p3.16xlarge` instance.

Other than the computational complexity of message passing we need to consider the (one-off) complexity pertaining the graph lifting procedures. Lifting procedures that are more likely to find immediate practical applications involve clique, cycle and induced cycle listing. For cliques, we refer readers to Bodnar et al. [8], where the authors report theoretical results regarding clique-listing complexity and the practical impact of employing specialised topological data analysis libraries.

As for cycle-based liftings, specialised cycle-listing algorithms exist. The algorithm in Birmelé et al. [7] is able to list all simple cycles in a graph in $\mathcal{O}(m + \sum_{c \in C(G)} |c|)$, where $m$ is the number of edges, $C(G)$ is the set of simple cycles in graph $G$ and $|c|$ is the size of the cycle. As for *induced*

cycles, the algorithm presented in Ferreira et al. [26] has a listing time of $\tilde{\mathcal{O}}(m + nC)$, with $n$ and $C$ being the number of nodes and induced cycles, respectively. In certain types of graphs, a better complexity can be obtained. In the case of planar graphs, Chiba and Nishizeki [16] show linear time complexity to list triangles and quadratic complexity for 4-rings. This is very important because almost all molecules are planar in a graph-theoretic sense [66] as a direct consequence of the chemical implications of Kuratowski's theorem [49]. However, we are not aware of any improved bounds for finding general induced cycles in planar graphs. Finally, we remind the reader that molecular rings can also be listed from the junction tree representation [27, 43], obtained by specialised molecular libraries such as RDKit [50].

Table 7: Wall-clock lifting times, mean and std over 5 runs (seconds).

| Dataset ↓ / Processes → | Seq. | 2 | 4 | 8 | 16 | 32 |
|---|---|---|---|---|---|---|
| ZINC (12k) | $320.27 \pm 0.54$ | $169.95 \pm 0.32$ | $84.90 \pm 0.21$ | $43.38 \pm 0.07$ | $23.17 \pm 0.68$ | $18.59 \pm 0.68$ |
| Mol-HIV (41k) | $1178.98 \pm 3.90$ | $635.58 \pm 0.83$ | $319.01 \pm 0.40$ | $164.26 \pm 0.52$ | $86.92 \pm 0.77$ | $60.62 \pm 2.05$ |
| ZINC-FULL (250k) | $6805.35 \pm 16.50$ | $3549.16 \pm 7.73$ | $1782.41 \pm 3.84$ | $918.38 \pm 3.46$ | $492.77 \pm 6.13$ | $383.92 \pm 3.30$ |

In our experiments, we implemented a lifting procedure based on the *generic* substructure matching algorithm exposed by the graph-tool Python library, which internally employs VF2 [19] to perform subgraph isomorphism. Noticing that the lifting procedure is embarrassingly parallel w.r.t. the independent graphs in a dataset, we easily parallelised the procedure via Python's Joblib library. On molecular benchmarks we observed the effective time required by preprocessing routines to always be modest compared to the training times. In Table 7 we report the wall clock runtimes, averaged over 5 runs, to lift all the graphs in the largest datasets amongst our benchmarks: ZINC, Mol-HIV and ZINC-FULL. The analysis has been conducted considering rings up to size 18 and by varying the number of parallel processing jobs on a server with an Intel® Xeon E5-2686 v4 processor with 64 vCPUs. It is possible to observe that the empirical lifting runtime scales linearly with the number of jobs in the range $[1, 16]$, and that such a simple parallelisation scheme dramatically reduces the preprocessing time on all datasets. When employing 32 parallel jobs, less than 19 seconds are required to preproceess the whole ZINC dataset, only 1 minute is required for Mol-HIV, and we needed slightly more than 6 minutes to lift all the 250k graphs in ZINC-FULL. We remark once more that these experiments have been conducted with a *generic* subgraph matching algorithm, and that even more parsimonious computation would be possible by using optimised ring-listing routines.

## C  Symmetries

In line with a recent effort in Geometric Deep Learning to understand different models through the lens of symmetry [12], we aim here to give a description of the underlying equivariance properties of CW Networks.

First, let us define the following matrix representation of the boundary relation from Definition 3.

**Definition 33.** *Let $X$ be a regular cell complex with $S_k$ denoting the number of cells in dimension $k$. The $k$-th unsigned boundary matrix $\boldsymbol{B}_k \in \mathbb{R}^{S_{k-1} \times S_k}$ of $X$ is given by $B_k(i, j) = 1$ if $\sigma_i \prec \sigma_j$ and $0$, otherwise.*

Let $X$ be a regular cell complex of dimension $n$ with boundary matrices $\mathbf{B} = (\boldsymbol{B}_1, \ldots, \boldsymbol{B}_n)$ and feature matrices $\mathbf{X} = (\boldsymbol{X}_0, \boldsymbol{X}_1, \ldots, \boldsymbol{X}_n)$ for the cells of different dimensions. Additionally, consider a sequence of permutation matrices $\mathbf{P} = (\boldsymbol{P}_0, \ldots, \boldsymbol{P}_n)$. Denote by $\mathbf{PX} = (\boldsymbol{P}_0 \boldsymbol{X}_0, \ldots, \boldsymbol{P}_n \boldsymbol{X}_n)$ and $\mathbf{PBP}^T = (\boldsymbol{P}_0 \boldsymbol{B}_1 \boldsymbol{P}_1^T, \ldots, \boldsymbol{P}_{n-1} \boldsymbol{B}_n \boldsymbol{P}_n^T)$.

**Definition 34.** *A function $f$ mapping $(\mathbf{X}, \mathbf{B}) \mapsto \mathbf{X}' = (\boldsymbol{X}_0', \ldots, \boldsymbol{X}_n')$ with the property that $\mathbf{P}f(\mathbf{X}, \mathbf{B}) = f(\mathbf{PX}, \mathbf{PBP}^T)$ for any $\mathbf{P}$ is called cell permutation equivariant.*

***Proof of Theorem 19.*** Definition 34 is similar to the (simplex) permutation equivariance definition from Bodnar et al. [8], with the subtle difference that the boundary matrices now have a more flexible structure in the case of cell complexes. The high-level idea is to see that all the adjacency matrices used by CWNs (i.e. $\boldsymbol{B}_k, \boldsymbol{B}_{k+1}, \boldsymbol{B}_k^\top \boldsymbol{B}_k, \boldsymbol{B}_{k+1} \boldsymbol{B}_{k+1}^\top$) are permuted accordingly by the permutation matrices in $\mathbf{P}$. Therefore, CWNs layers computes the same function up to a permutation of the cells. The proof follows a similar logic to to the one in Bodnar et al. [8] for simplicial networks, and we refer the reader to their work for a detailed proof. $\qquad\square$

It is common in algebraic topology and differential geometry to equip the incidence relation $\sigma \prec \tau$ with additional structure that makes it a signed incidence relation.

**Definition 35** (Hansen and Ghrist [36])**.** *A signed incidence relation on $P_X$ is a map $[\cdot : \cdot] \colon P_X \times P_X \to \{0, \pm 1\}$ with the properties:*

1. *If $[\sigma : \tau] \neq 0$, then $\sigma \prec \tau$.*

2. *For any $\sigma \leq \tau$, $\sum_{\gamma \in P_x} [\sigma : \gamma][\gamma : \tau] = 0$.*

This signed incidence relation can be be encoded by the signed incidence (boundary) matrices of $X$. We define these below:

**Definition 36.** *Let $X$ be a regular cell complex with a signed incidence relation $[\cdot : \cdot]$. Let $S_k$ denote the number of cells in dimension $k$. The $k$-th signed boundary matrix $\boldsymbol{B}_k \in \mathbb{R}^{S_{k-1} \times S_k}$ of $X$ is given by $B_k(i, j) = [\sigma_i : \sigma_j]$.*

The difference with respect to the unsigned boundary matrices is that the non-zero values of the matrix can be $\pm 1$, not just 1. This can be used to define a notion of orientation equivariance for CW Networks. This ensures that when changing the orientation of the cell complex $X$ (i.e. changing $[\cdot : \cdot]$) one computes the same function up to that change in orientation.

Let $X$ be a regular cell complex of dimension $n$ described by the *signed* boundary matrices $\mathbf{B} = (\boldsymbol{B}_1, \ldots, \boldsymbol{B}_n)$ and feature matrices $\mathbf{X} = (\boldsymbol{X}_0, \boldsymbol{X}_1, \ldots, \boldsymbol{X}_n)$ for the cells of different dimensions. Additionally, consider a sequence of diagonal matrices $\mathbf{T} = (\boldsymbol{T}_0, \ldots, \boldsymbol{T}_n)$ with values in $\pm 1$. Additionally, let $\boldsymbol{T}_0 = \boldsymbol{I}$. Denote by $\mathbf{TX} = (\boldsymbol{T}_0 \boldsymbol{X}_0, \ldots, \boldsymbol{T}_n \boldsymbol{X}_n)$ and $\mathbf{TBT} = (\boldsymbol{T}_0 \boldsymbol{B}_1 \boldsymbol{T}_1, \ldots, \boldsymbol{T}_{n-1} \boldsymbol{B}_n \boldsymbol{T}_n)$.

**Definition 37.** *A function $f$ mapping $(\mathbf{X}, \mathbf{B}) \mapsto \mathbf{X}' = (\boldsymbol{X}'_0, \ldots, \boldsymbol{X}'_n)$ with the property that $\mathbf{T} f(\mathbf{X}, \mathbf{B}) = f(\mathbf{TX}, \mathbf{TBT})$ for any $\mathbf{T}$ is called orientation equivariant.*

Making CWNs orientation equivariant requires imposing additional constraints on the layers of the model. This proceeds similarly to MPSNs [8]. Since applications involving oriented simplicial complexes are out of the scope of this work, we refer the reader to Bodnar et al. [8] for an intuition of how this can be extended to cell complexes.

## D  Sheaves, Laplacians and Convolutions

It is useful on cell complexes to derive a Laplacian operator based on cellular sheaves [36], since many interesting Laplacians, such as the (normalised) graph Laplacian [17], the Hodge Laplacian [63] and the connection Laplacian [67] can be obtained as particular cases. Intuitively, a cellular sheaf is a construction that assigns a vector space to each cell in the complex and a (linear) map for each face relation in the complex $\sigma \leq \tau$. Additionally, these linear maps must satisfy some compositionality constraints imposed by the structure of $P_X$.

### D.1  Sheaf Laplacian

**Definition 38.** *Let $(X, P_X)$ be a regular cell complex, and denote by $\mathrm{Hilb}_K$ the class of Hilbert spaces over a field $K$. A **weighted cellular sheaf** $\mathcal{F}$ is given by the assignment*

$$\mathcal{F} \colon P_X \to \mathrm{Hilb}_K$$
$$\sigma \mapsto \mathcal{F}(\sigma)$$

*together with a bounded linear map $\mathcal{F}_{\sigma \leq \tau} \colon \mathcal{F}(\sigma) \to \mathcal{F}(\tau)$ for any $\sigma \leq \tau$.*

*This data satisfies that $\mathcal{F}_{\sigma \leq \sigma} = id$ for all $\sigma \in P_X$ and $\mathcal{F}_{\sigma \leq \omega} = \mathcal{F}_{\tau \leq \omega} \circ \mathcal{F}_{\sigma \leq \tau}$ whenever $\sigma \leq \tau \leq \omega$.*

Given a weighted cellular sheaf $\mathcal{F}$, we define a chain complex as follows. For each $k = 0, 1, \ldots$ we set

$$C^k(X; \mathcal{F}) = \bigoplus_{dim(\sigma) = k} \mathcal{F}(\sigma).$$

Further, we define coboundary maps $\delta^k \colon C^k(X; \mathcal{F}) \to C^{k+1}(X; \mathcal{F})$ by

$$\delta^k(x)_\tau = \sum_{dim(\sigma) = k} [\sigma : \tau] \mathcal{F}_{\sigma \leq \tau}(x_\sigma),$$

where $[\cdot : \cdot] \colon P_X \times P_X \to \{0, \pm 1\}$ is a signed incidence relation (see Definition 35).

Given Hilbert spaces $V$ and $W$ and a bounded linear map $T \colon V \to W$, the adjoint of $T$ is the unique bounded linear map $T^\star \colon W \to V$ satisfying that for all $v \in V$ and all $w \in W$:

$$\langle w, Tv \rangle = \langle T^\star w, v \rangle .$$

**Definition 39.** *Let* $C^\bullet = C^0 \to C^1 \to \dots$ *be a chain complex of Hilbert spaces. The* **Hodge Laplacian** *is the graded linear map defined in degree* $k$ *as* $\Delta^k \colon C^k \to C^k$ *with* $\Delta^k = (\delta^k)^\star \delta^k + \delta^{k-1}(\delta^{k-1})^\star$. *When* $C^\bullet = C^0 \to C^1 \to \dots$ *is the complex of cochains of a weighted cellular sheaf* $\mathcal{F}$, *the Hodge Laplacian is called the* **sheaf Laplacian** *of* $X$.

In particular, the Hodge Laplacian of a cell complex can be obtained by considering the *constant weighted cellular sheaf* with a standard inner product. That is the cellular sheaf where $\mathcal{F}(\sigma) = \mathbb{R}$ and the restriction maps $\mathcal{F}_{\sigma \leq \tau} = \mathrm{id}$. A normalised version of it can also be obtained by carefully adjusting the inner products associated with each $\mathcal{F}(\sigma)$. This normalisation is always possible for finite cell complexes (see Hansen and Ghrist [36] for details). This is very useful because finding normalised versions of Hodge Laplacians is not trivial and even on simplicial complexes [63], the process of constructing one can be quite involved.

## D.2 Convolutional Operators

One can use the general sheaf Laplacian to define linear, local diffusion operators which, in the GNN literature, are broadly addressed as 'convolutional'. Diffusion operators built from the standard graph Laplacian have been employed in several graph neural network architectures [22, 47]. Recent works [13, 25] have introduced convolutional operators on SCs by employing the Hodge Laplacian [63], interpreted as a generalisation of the graph Laplacian. As for cell complexes, here we focus, for simplicity, on the case of a constant sheaf with a standard inner product in $\mathbb{R}^n$. Then, the matrix representations of $\delta^k$ and $(\delta^k)^*$ are the signed incidence matrices $\boldsymbol{B}_k^T$ and $\boldsymbol{B}_k$, respectively. Therefore, the Hodge Laplacian can be written in matrix form as

$$\boldsymbol{L}_k = \boldsymbol{B}_k^T \boldsymbol{B}_k + \boldsymbol{B}_{k+1} \boldsymbol{B}_{k+1}^T .$$

A convenient way to define a convolutional operator on cochains is by designing a learnable filter parameterised as a polynomial of the Hodge Laplacian. This approach has been already adopted on graphs using the standard graph Laplacian [22] or more general sheaf Laplacians [37], and also on simplicial complexes [25]. The advantage of this approach is that of retaining a connection with spectral constructions [22, 25] while not requiring any explicit diagonalisation of the operator itself. A polynomial convolutional filter of this kind, when applied to the $p$-cells of a $d$-cell complex, would take the form

$$H^{t+1} = \psi \Big( \sum_{r=0}^{R} \boldsymbol{L}_p^r H^t W_r^{t+1} \Big) = \psi \Big( H^t W_0^{t+1} + \sum_{r=1}^{R} \boldsymbol{L}_p^r H^t W_r^{t+1} \Big). \tag{3}$$

where $H^t$ is a matrix gathering $p$-cell representations at layer $t$, $W_r^{t+1}$ are learnable parameters, and $\psi$ summarises the application of a bias term and a non-linearity.

***Proof of Theorem 19.*** While the structure of the boundary matrices is more flexible in a cell complex than in a simplicial complex, algebraically, the proof is very similar to the proof showing MPSNs generalise simplicial convolutions in Bodnar et al. [8]. We offer here a high-level view of the proof and refer the reader to Appendix C of their paper for a detailed version.

For a generic $p$-cell $\sigma$, and $r > 0$, the application of the $r$-power of the Hodge Laplacian effectively induces an information flow from a generalised notion of $r$-upper and $r$-lower adjacent $p$-cells, i.e. $p$-cells $\tau$ such that there exists a sequence of upper- (respectively, lower-) adjacent $p$-cells $[\gamma_0, \gamma_1, \dots, \gamma_r]$ such that $\gamma_0 = \sigma, \gamma_r = \tau$.

Therefore, the convolution described above is easily interpreted in terms of a cellular message passing scheme which only exchanges ↑- and ↓-messages. Intuitively, the upper- and lower- message functions would share their parameters $W_r^{t+1}$ and compute messages by linearly projecting the representations of upper- and lower-adjacent cells (ignoring any information in shared (co)boudaries). Such messages would then be aggregated by summation into an overall message, taken as input by an update function

parameterised by $\psi$ and $W_0^{t+1}$. A formal derivation of how the equation (3) is rewritten in terms of cellular message passing would closely follow the one provided in Bodnar et al. [8] for SCs, and we therefore refer readers to Section C of such work. $\qquad\square$

Normalised versions of the aforementioned Hodge Laplacian can be used to design a model in the spirit of the popular Graph Convolutional Network of Kipf and Welling [47]. To this aim, one could resort to normalised sheaves as suggested in [36]. Additionally, one could explicitly make use of the (co)boundary operators defined in Section C to let information flow from lower- and higher-dimensional cells contained in cell (co)boundaries, effectively extending the Simplicial Convolutional Networks recently introduced in Bunch et al. [13]. We defer these research directions to future developments of this work.

# E  Experimental details and additional results

## E.1  Used Code Assets

The model has been implemented in PyTorch [59] and by building on top of the PyTorch Geometric library [28]. Lifting operations use the graph-tool[4] Python library and are parallelised via Joblib[5]. PyTorch, NumPy, SciPy and Joblib are made available under the BSD license, Matplotlib under the PSF license, graph-tool under the GNU LGPL v3 license. PyTorch Geometric is made available under the MIT license.

## E.2  Used Computer Resources

All experiments were run on NVIDIA® GPUs. Experiments on **SR**, **Mol-HIV** and molecular **TUDatasets** were run on Tesla V100 GPUs with 5,120 CUDA cores and 16GB GPU memory on a `p3.16xlarge` Amazon Web Services (AWS) Elastic Cloud (EC) 2 instance. Experiments on the social **TUDatasets** were run on the same GPU devices but with 32GB HBM2 memory mounted on an HPC cluster. All remaining experiments, that is **CSL**, **RingTransfer** and **ZINC**, were run on a machine with TITAN Xp GPUs with 12GB GPU memory and an Intel® Xeon® CPU E5-2630 v4 @ 2.20GHz CPU.

## E.3  Model

In all cases, we apply our model to the 2-dimensional cell complexes obtained by ring-lifting the original graphs, i.e. we consider nodes and edges as 0- and 1-cells, and each induced cycle of size up to $k$ as a 2-cell. 0-cell are always endowed with the original node features or learnt node embeddings, if the benchmark prescribes so. The way higher dimensional cells are assigned features depend on the specific benchmark.

Throughout all experiments, we employ cellular message passing layers which update the representation of $p$-cell $\sigma$ as follows:

$$h_\sigma^{t+1} = \mathrm{MLP}_{U,p}^t\Big(\mathrm{MLP}_{\mathcal{B},p}^t\big((1+\epsilon_\mathcal{B})h_\sigma^t + \sum_{\tau\in\mathcal{B}(\sigma)} h_\tau^t\big) \parallel$$

$$\mathrm{MLP}_{\uparrow,p}^t\big((1+\epsilon_\uparrow)h_\sigma^t + \sum_{\tau\in\mathcal{N}_\uparrow(\sigma),\delta\in\mathcal{C}(\sigma,\tau)} \mathrm{MLP}_{M,p}^t\big(h_\tau^t \parallel h_\delta^t\big)\big)\Big) \qquad (4)$$

Here, $\parallel$ indicates concatenation, $\mathrm{MLP}_{\mathcal{B},p}^t$, $\mathrm{MLP}_{\uparrow,p}^t$ are 2-Layer Perceptrons and $\mathrm{MLP}_{U,p}^t$, $\mathrm{MLP}_{M,p}^t$ consist of a dense layer followed by a non-linearity. We neglect messages from cofaces and down-adjacent cells consistently with Theorem 7. We name an architecture which stacks $L$ layers of this form as 'Cell Isomorphism Network' (CIN). Readout operations are performed as follows. First, for $p \in 0, 1, 2$, we compute the joint representation $h_p$ of the cells at dimension $p$ by applying a mean or sum readout operation. Then, for complex $\mathcal{K}$, we compute an overall representation $h_\mathcal{K} = \sum_{p=0,1,2} \mathrm{MLP}_{R,p}(h_p)$, where each $\mathrm{MLP}_{R,p}$ is parameterised as a single dense layer followed

---

[4] https://graph-tool.skewed.de/
[5] https://joblib.readthedocs.io/en/latest/

by a non-linearity. Complex-wise predictions are obtained by a final dense layer preceded by dropout. All MLP layers internally apply Batch Normalization [42] and ReLU activations, unless otherwise specified. All training procedures are performed with the Adam optimiser [46].

### E.4    Additional experimental details

**CSL**    Each of the 150 4-regular graphs in the CSL dataset comprises $N = 48$ nodes and is characterized by *skip number* parameter $C \in \mathcal{C} = \{2, 3, 4, 5, 6, 9, 11, 12, 13, 16\}$. Parameters $N$ and $C$ determine the isomorphism class $\mathcal{G}_{N,C}$ of each graph, which we seek to predict. The number of possible classes is $|\mathcal{C}| = 10$. We employ the same stratified dataset folds in Dwivedi et al. [24]. Consistently with the adopted reference procedure, 0-cells share the same learnt embedding, while 1- and 2-cells are endowed with the sum of the embeddings of the included 0-cells. As for the optimisation procedure, we set the batch size to 12 and the initial learning rate of 5e-4. which is halved whenever the validation performance does not improve after a patience value of 20. The training is early stopped as soon as it falls below 1e-6, at which step we measure the model test accuracy. The size of hidden layers in our model is set to 160 and we stack 3 cellular message passing layers. In this benchmark, we replace Batch Normalisation with Layer Normalization [5], as the former wsa observed to produce instabilities in the optimisation procedure. At each dimension, cell embeddings are readout via averaging.

**SR**    These experiments are run in double floating point precision and with untrained models. We initialise the cell complexes associated with SR graph by populating 0-cells with constant, scalar, unitary signal, and 1- and 2-dimensional cells with the sum of the contained 0-cells. Complexes are embedded in a 16-dimensional space and, coherently with Bodnar et al. [8] and Bouritsas et al. [10], if the $L_2$-distance between the embeddings of two complexes is larger than $\epsilon = 0.01$, we deem the corresponding graphs to be non-isomorphic. We confirmed the validity of the chosen threshold $\epsilon$ by numerically verifying that, under the described experimental setting, each SR graph in our datasets is deemed isomorphic w.r.t. a counterpart obtained by randomly permuting its nodes. We run a CIN model with 3 cellular message passing layers, whose hidden layers comprise 16 units. At each dimension, cell embeddings are readout via summation. As the number of induced cycles of a certain size may be enough to tell apart non-isomorphic SR graphs (see Table 4), an MLP with sum readouts represents a strong baseline, which we additionally run. Such a model applies non-linear dense layers at each cell dimension, and then performs readout operations as in CIN. We set the size of hidden layers to 256, while the final complex embeddings are embedded in a 16-dimensional space as in our model. Both approaches are equipped with ELU nonlinearities [18].

**RingTransfer**    This benchmark dataset comprises 5, 000 training graphs. Each graph is randomly associated with one of the 5 independent labels, which are also assigned as node features to **source** nodes. Labels are unifomly represented. On this benchmark we run a CIN model with 3 stacked message passing layers, independently on the ring size. The hidden size of the layers is set to 64 and we do not apply Batch Normalisation. Differently than in the other benchmarks, we do not need to perform readout operations to compute complex-wise embeddings; instead, we simply take the representation of the 0-cell corresponding to node **target** at the last layer of the architecture and use it to predict the label of **source**. GIN models have always $\lfloor \frac{k}{2} \rfloor$ standard message passing layers with hidden size 64. The models are trained with an initial learning rate of $10^{-3}$, decayed by a factor of 0.5 and a patience of 5 epochs. The training is stopped when the learning rate drops below $10^{-5}$.

**TUD**    Amongst the datasets from this benchmarking suite: the task in **MUTAG** is to recognise mutagenic molecular compounds for potentially marketable drug [45, 61]; the one in **PTC** is to recognise the chemical compounds according to carcinogenicity on rodents [39, 48]; **PROTEINS** is about to categorising proteins into enzyme and non-enzyme structures [9, 23]; **NCI1** and **NCI109** deal with identifying chemical compounds against the activity of non-small lung cancer and ovarian cancer cells, respectively [70]; **REDDIT-BINARY** or **RDT-B** is a social network dataset where the task is to predict whether a graph belongs to a question-answer-based community or a discussion-based community. On these datasets, we followed the approach in Xu et al. [74], which prescribes to run a 10-fold cross-validation procedure and report the maximum of the average validation accuracy across folds. Consistently with such work, we train our model starting from an initial learning rate which is decayed after a fixed amount of epochs and we apply cell-readout operations on the multiscale representations obtained by a Jumping Knowledge scheme [73] by performing averaging

Table 8: Hyperparameter configurations on TUDatasets.

| Hyperparameter | MUTAG | PTC | PROTEINS | NCI1 | NCI109 | IMDB-B | IMDB-M | RDT-B |
|---|---|---|---|---|---|---|---|---|
| Batch Size | 32 | 32 | 128 | 32 | 32 | 128 | 128 | 32 |
| Initial LR | 0.01 | 0.01 | 0.01 | 0.001 | 0.001 | 0.001 | 0.0005 | 0.001 |
| LR Dec. Steps | 20 | 50 | 20 | 20 | 20 | 50 | 20 | 50 |
| LR Dec. Strength | 0.5 | 0.9 | 0.5 | 0.5 | 0.5 | 0.5 | 0.5 | 0.5 |
| Hidden Dim. | 64 | 16 | 32 | 16 | 64 | 16 | 64 | 64 |
| Drop. Rate | 0.5 | 0.0 | 0.0 | 0.5 | 0.0 | 0.0 | 0.5 | 0.0 |
| Drop. Pos. | cell read. | comp read. | comp read. | comp read. | comp read. | comp read. | comp read. | comp read. |
| Initialisation | sum | mean | mean | mean | mean | mean | mean | mean |
| Cobound. in $\uparrow$-msg | N | N | Y | Y | Y | N | N | N |
| Num. Layers | 4 | 4 | 3 | 4 | 4 | 4 | 4 | 4 |

or summation depending on the dataset, still in accordance with Xu et al. [74]. We ran a grid-search to tune batch size, hidden dimension, dropout rate, initial learning rate along with its decay steps and strengths, feature initialisation strategy of higher-dimensional cells (mean vs. sum), inclusion of coboundary features in $\uparrow$-messages, number of layers and the dropout position (immediately after readout on cells ("cell read.") or the final readout on the complex ("comp read.")). We report the hyperparameter configurations in Table 8. We finally report that we did not employ Batch Normalization layers in **RDT-B** since they were observed to produce severe instabilities in the training procedure.

**ZINC** The ZINC benchmarks dataset have been constructed by the ZINC database provided by the Irwin and Shoichet Laboratories in the Department of Pharmaceutical Chemistry at the University of California, San Francisco (UCSF) [68]. Each graph represents a molecule, with node features indicating the atom type and edge features the type of chemical bond between two atoms. Graph targets correspond to the penalised water-octanol partition coefficient – logP [33]. In these experiments, rings up to size $k = 18$ are mapped to 2-cells, and are assigned feature values as the sum of the learnable atom embeddings for the included 0-cells (nodes). 1-cells are assigned learnable bond embeddings if edge-features are considered, otherwise we apply the same policy employed for 2-cells. We employ the same predefined training, validation and test splits as in Dwivedi et al. [24], and train our model by minimising the the Mean Absolute Error (MAE) loss on the train targets. As prescribed by the benchmark, the optimisation procedure employs a batch size of 128 and a dynamic learning rate which starts from $10^{-3}$ and is halved whenever the validation loss does not improve after a patience value we set to 20. The training is early stopped as soon as it falls below $10^{-5}$. We repeat the training with 10 different weight initialisations and report the mean of the test MAEs at the time of early stopping. In accordance with the best performing baselines, our CIN model does not use any dropout, and stacks 4 message passing layers with hidden size 128. In order to enforce the parameter budget we reduce the size of hidden layers to 48 and only perform 2 message passing layers. At each dimension, cell embeddings are readout via summation.

**Mol-HIV** This dataset comprises 41127 molecular graphs associated with a binary label representing their capacity to inhibit HIV replication. The benchmark provides predefined train, validation and test sets based on the "scaffold splitting" procedure, which separates molecules based on their two-dimensional structural frameworks [40]. As in **ZINC**, graphs are attributed at the level of nodes and edges, and we directly employ the atom and bond embedding layers provided by the benchmarking platform[6] to populate 0- and 1-dimensional cells. Rings of size up to $k = 6$ are considered as 2-cells, and are endowed with feature vectors with the same procedure as in **ZINC**. The value $k = 6$ has been chosen from the pool of values $\{6, 8, 18\}$ as it yielded the highest validation performance. The architecture hyperparameters are directly replicated from the HIMP model in Fey et al. [27]: 2 message passing layers, dropout rate of 0.5 applied after each layer, 64 as size of hidden layers, constant learning rate of $10^{-4}$, batch size of 128. We train our model for 150 epochs. The small CIN model is obtained by simply reducing the size of hidden layers to 48. At each dimension, cell embeddings are readout via averaging.

---

[6] https://github.com/snap-stanford/ogb/blob/master/ogb/graphproppred/mol_encoder.py

## E.5 Ablation study on ZINC

We end this section by reporting the results of an ablation study we conducted on the ZINC dataset to appreciate the contribution of including rings. In Table 9 we show the average test MAE for two additional CIN models: "CIN No-Rings small" and "CIN No-Rings", which differ from their original counterparts in that they neglect 2-cells when performing message passing. In these experiments we always make use of edge features and use the same hyperparameters as our original CIN model.

Table 9: ZINC Ablation with edge features. The ablation shows the benefits of integrating rings into the message passing procedure.

| Method | MAE |
| --- | --- |
| GatedGCN [11] | 0.363±0.009 |
| GIN [74] | 0.252±0.014 |
| PNA [20] | 0.188±0.004 |
| DGN [6] | 0.168±0.003 |
| HIMP [27] | 0.151±0.006 |
| GSN [10] | 0.108±0.018 |
| GIN-E Custom | 0.196±0.007 |
| CIN No-Rings small | 0.174±0.006 |
| CIN No-Rings | 0.159±0.007 |
| CIN-small | 0.094±0.004 |
| CIN | **0.079±0.006** |

In line with our expectations, we observe a decrease in the overall performance of both versions. They are outperformed by the GSN [10] and HIMP [27] models, which either include structural information from cycle isomorphism counting (GSN) or additionally perform message passing on the Junction Tree representation of molecules (where rings are considered as nodes). At the same time, we observe "CIN No-Rings" still outperforms all other ring-agnostic baselines. We attribute such strong performance to the more natural and richer modelling of edge signals (1-cells): this model updates edge representations at each layer as a function of the present representations and those of the incident nodes (0-cells). As an additional confirmation of this hypothesis, we implemented an architecture which replicates the same structure as "CIN No-Rings", but replaces cellular message passing with GIN-E layers [41]. These layers extend the message passing scheme in GIN by accounting for edge features. We refer to this model as "GIN-E Custom". Contrary to CIN, it does not update edge representations and performs readout only at the node level. As expected, we observed that "GIN-E Custom" is outperformed by all our models, including, in particular, "CIN No-Rings".