# OpenReview forum: "Weisfeiler and Lehman Go Cellular: CW Networks"
_NeurIPS.cc/2021/Conference — NeurIPS 2021 Poster_

### Official Review · Reviewer_1iwy · 2021-07-08

**Rating:** 8
**Confidence:** 3

**Summary:**

In this work, the authors propose a new member of graph neural networks, called CWN, that considers the higher-order structure of graphs. The proposed model leverages the topological properties of cell complexes and establishes a novel hierarchical message passing procedure. The authors demonstrate that the proposed CWN is more powerful than the WL test and can be comparable to the 3-WL test. The proposed model achieves encouraging performance on graphs with complicated structures, e.g., molecules, while it requires a preprocessing step with the tolerable time cost.

**Limitations And Societal Impact:**

I did not find any obvious or potential negative societal impact of the proposed work.

**Main Review:**

Overall, I like this work because of its solid theoretical analysis, sufficient and convincing experimental results, and detailed supplementary materials. Additionally, in Section 4, the authors connect the proposed method with existing works and highlight its advantages, which further clarifies the contribution of this work to the community.

I only have four suggestions:

1. Line 121: 	“Simplicial WL” -> “Simplicial WL (SWL)”, such that SWL will be understandable.

2. The GWL coloring procedure (the preprocessing step) shown in Figure 3 provides a hierarchical way to extract features from graphs. If my understanding is correct, we can define a graph kernel based on the features and provide a kind of high-order WL kernel accordingly. Could the authors take this potential kernel method as a baseline in the experiment?

3. The introduction of CWN in Section 4 is too brief. It would be nice if the authors can illustrate the molecular message-passing model by a figure — after doing the coloring procedure, how to apply message passing to the coloring result? Maybe the authors can consider using a simple molecule as the graph in Figure 3 and adding a subfigure to it.

4. Reading this paper is challenging for the readers without topological background. Section 2 is not a good starting point for such readers. Before showing Figure 1, it would be necessary to add a Figure to illustrate topological space X, its partitions, and cells, with sufficient symbols and notations, such that it will be much easier to understand the concepts in definition 1.

**Time Spent Reviewing:**

6

---

> ### Author Response · Authors · 2021-08-09
> **Official response to Reviewer 1iwy**
>
> We are happy to note the reviewer has appreciated the solidity of the theoretical analyses and empirical evaluations. We respond to all the outstanding comments and questions below.
>
> __A cellular WL kernel method as a baseline__
>
> In general, any cell complex kernel paired with a cellular lifting map produces a valid graph kernel. To the best of our knowledge, cell complex kernels have not been studied in the literature and we agree this would be an interesting line of future work. In particular, based on our results, a CWL-based kernel could be a strong method on the molecular benchmarks we study and we agree it would represent a sensible baseline. We will try to implement and compare with such a baseline for the final version of the paper.
>
> __“Reading this paper is challenging for the readers without a topological background. Section 2 is not a good starting point for such readers. Before showing Figure 1, it would be necessary to add a Figure to illustrate topological space X, its partitions, and cells, with sufficient symbols and notations”__
>
> We agree with the reviewer on adding an illustrative figure to accompany the definition of a cell complex. We also plan to use an alternative definition that more closely matches the intuition of hierarchical gluing operations that is given in the main text. We will make use of the additional space available for the final version to make these changes and improve the overall clarity of Section 2.
>
> __“The introduction of CWN in Section 4 is too brief. It would be nice if the authors can illustrate the molecular message-passing model by a figure”__
>
> We agree. We will use the additional space for the final version to add such a figure. Thank you for the suggestion!
>
> __“ “Simplicial WL” -> “Simplicial WL (SWL)”, such that SWL will be understandable ”__
>
> We will add. Thank you!

---

> > ### Comment · Reviewer_1iwy · 2021-09-03
> > **After rebuttal**
> >
> > After reading other reviewers' comments and the authors' replies, I still believe that this is an interesting work, so I keep my score (8) unchanged. Thank the authors for their detailed explanations.

---

> > > ### Author Response · Authors · 2021-09-03
> > > **Response**
> > >
> > > Thank you for your positive feedback and helpful suggestions during the review process!

---

### Official Review · Reviewer_hkAr · 2021-07-16

**Rating:** 6
**Confidence:** 4

**Summary:**

Recently, there have been several attempts to put higher-order structures on graphs and perform message passing over these structures. One example is the Message Passing Simplicial Networks proposed in e.g, ref[4]. The present paper extends the higher-order structures from simplicial complexes (SCs) to regular cell complexes (also known as CW complexes), or more precisely, generalizes the Simplicial WL of ref[4] to a Cellular WL (CWL). Both theoretical understanding, and experimental studies of CWL are presented. Specific contributions include:

(1) To obtain a CW complex from a graph, the paper introduces a ``cellular lifting map", and shows a skeleton-preserving lifting map gives rise to a CWL that is at least as powerful as standard WL (Weisfeiler-Lehman) in graph isomorphism testing. It also gives examples of specific natural lifting maps (which includes the previous SWL in [4] as a special case) that is at least as powerful as the so-called 3-WL.

(2) It describes and implements a CW network using a lifting map that attaches 2-cells to all induced cycles in the input graphs. This architecture (CWN) makes sense especially for molecular graphs applications (in capturing both the covalent bonds and chemical rings).

(3) It presents various experimental results on synthetic and real datasets, which show that a specific instantiation of CWN, called CIN, can both differentiate certain hard cases for graph isomorphism testing (e.g, 4-regular graphs), comparable or better performance on TUDatasets, and better performance in molecular benchmark datasets.


**Limitations And Societal Impact:**

Adequate

**Main Review:**

I want to thank the authors for their responses to my questions in the review and during the discussion period. Please add a discussion/clarification of Theorem 15 as discussed. I maintain my score of the paper.

See the above for the summary of the paper and its contribution.

** strength of the paper:
(1) Graph neural networks (especially message passing graph networks) have received great attention in recent years. Recently there has been much interest in message passing over higher order structures, mostly to increase the expressiveness power of graph networks. Using CW complexes as in this paper is more flexible (e.g, filling induced cycles by 2-cells) than using simplicial complexes. The use of CWL is both natural and useful in practical applications.

(2) The authors present several theoretical results on CWL and CWN. The formulation of lifting map is interesting.

(3) Good empirical studies are presented, with both synthetic datasets to show the expressiveness and real benchmark datasets to see performance improvements.


** weakness of the paper:
(a) The development of CWL is a generalization of the simplicial version (SWL, ref[4]). Most of the theoretical results feel incremental compared to the work of ref[4]. (The notion of cellular lifting map is new but perhaps not surprising. I have read the supplement briefly as well, and understand that there are complications / technicality that need to be addressed due to the use of cells instead of simplices. The concept of cellular message passing also exists in ref[24].) If there are significant new ideas, it would be great if they can be articulated.
Furthermore, just being more expressive than WL is not a very strong theoretical result, especially since CWL is using high-dimensional cells.

(b) (A minor one:) I like the discussions and theoretical results on expressive power in distinguishing non-isomorphic graphs. There are other ways to increase expressive power, for example, by using more powerful initial node features (say sth. that captures cycle information) or by using node coloring (e.g, the following reference):
      [Ref]: "Coloring Graph Neural Networks for Node Disambiguation",
      George Dasoulas, Ludovic Dos Santos, Kevin Scaman, Aladin Virmaux,
      IJCAI 2020.
It could be good to include a comparison with the approach in the above Ref.

** overall evaluation:
Overall, I think this is a solid paper that presented nice studies over a natural object with reasonably convincing experimental results (especially over molecular graphs). The originality is somewhat limited (e.g., when compared to work of Ref[4]). Nevertheless, it is well-written and I enjoyed reading it.

** other comments:
-- Maybe I missed it in the submission and supplement: how is the ring size $k$ chosen in experiments (especially for real datasets)? Different values of $k$ seem to be used for different applications. Is there a consistent strategy to choose it?
There is a range of hyperparameters listed in Table 8 in supplement. Why not having $k$ as another hyperparameter? Also, how does the number of hyperparameters compare to other standard approaches, say GIN?
-- In Definition 1, is "regularity" important for the CW network? Is it needed for any of the results later?

**Time Spent Reviewing:**

6 hours

---

> ### Author Response · Authors · 2021-08-09
> **Official response to Reviewer hkAr**
>
> We are glad to see the reviewer has appreciated the usefulness of the proposed model in practical applications and has found our theoretical analysis interesting. We have responded below to all the outstanding questions and suggestions.
>
> __Relationship to the works in [4, 24] and articulating the significant new ideas__
>
> Although a cellular message passing scheme was already introduced in ref. [24], our work brings significant additional contributions in studying the expressive power of such schemes along with a theoretically grounded framework to apply them on graph structured data in a way to address several limitations of standard Graph Neural Networks. These contributions find extensive empirical validation in our work, one additional aspect lacking in [24]. We also remark that the message passing formulations differ in the two papers. The specific combination of adjacencies used in our paper is theory-driven. Our scheme is a direct result of Theorem 7, which we generalise from [4], and allows us to perform message passing efficiently without sacrificing the expressive power of the model. In contrast, the schemes proposed in [24] either suffer from very high-computational complexity or their expressive power is unknown / not studied.
>
> In relation to ref. [4], other than preseting new theoretical results in relation to the model proposed in such work (see next paragraph), we deepen and generalise the analysis on lifting transformations in a principled way to CW complexes. Importantly, this unlocks applications of higher-order message passing to domains such as molecular modelling – for which clique-liftings would not be appropriate, since higher-order cliques are not present. Also, this indirectly addresses other limitations of standard GNNs such as over-squashing or over-smoothing. We will articulate more the differences with respect to these works as suggested in the final version.
>
> __“Just being more expressive than WL is not a very strong theoretical result, especially since CWL is using high-dimensional cells.”__
>
> We note that the WL results are only a fraction of the results we present in the paper, many of which are included only in the appendix due to space limitations (we will better signpost these in the final version). In particular, we also include results comparing CWNs / CWL to MPSNs / SWL. All the expressivity results concerning CWNs are summarised in Corollary 27 (Appendix A) which we include in full below:
> >CWNs have the following properties:
> >1. They are at least as powerful as the WL test when using skeleton-preserving lifting transformations.
> >2. They are strictly more powerful than the WL test when using the lifting maps from Corollary 14.
> >3. They are not less powerful than 3-WL when using the lifting transformations from Theorem 15.
> >4. They are at least as powerful as MPSNs [7] when using a lifting transformation whose output complex contains the clique complex as a subcomplex.
> >5. They are strictly more powerful than MPSNs when using a lifting transformation based on rings/cycles and the clique complex of the graph. In particular, CWN using rings is strictly more powerful than MPSNs using a lifting based on triangles, since triangles are rings of size 3.
>
> Besides these results, it is also worth mentioning the non-trivial result from Theorem 7, which enables us to perform message passing on cell complexes efficiently without sacrificing the expressive power of the model.
>
> __Other ways to improve expressive power (e.g. _Coloring Graph Neural Networks for Node Disambiguation_)__
>
> The expressive power of GNNs is a topic which has attracted particular attention in recent years. Several methods have been proposed including approaches which augment nodes with additional information of various kinds. One such approach is represented, in particular, by GSN (ref. [9]), which endows nodes with isomorphism counting of substructures. Regular message passing is, however, run at the level of nodes. This model has proved to be particularly effective on molecular benchmarks when considering induced cycles in the subgraph bank, and we have thus chosen it as a strong baseline in many of our experiments. We significantly outperform this method in our molecular experiments and, often, in the smaller TU datasets. We will better signal the importance of this baseline in the paper.
>
> The reported reference _Coloring Graph Neural Networks for Node Disambiguation_ represents indeed another approach to improve the expressive power of GNNs via node feature augmentation, but, contrary to the aforementioned methods, it does not feature an inductive bias specifically oriented to molecular modelling. We will, however, introduce it and discuss the differences w.r.t. our approach in the paragraph devoted to related works.
>
> __“How is the ring size k chosen in experiments (especially for real datasets)? Different values of k seem to be used for different applications. Is there a consistent strategy to choose it? There is a range of hyperparameters listed in Table 8 in supplement. Why not having k as another hyperparameter?”__
>
> The maximum size of rings in the lifting procedure ($k$) is indeed an additional hyperparameter. We will make sure to clarify this aspect in the text. Any standard hyperparameter tuning strategy can be employed for the choice of this value, which can also potentially be driven by domain knowledge on the task at hand. As for ZINC, we simply choose $k$ to be the maximum induced cycle size in the training dataset. In MOLHIV, we chose the value yielding the highest validation performance from a small set of values (6, 8, 12). In TU datasets, we did not tune this value. Setting it to 6 (typically the most represented in molecular datasets) sufficed to attain good performance.
>
> __“How does the number of hyperparameters compare to other standard approaches, say GIN?”__
>
> In general, the hyperparameter overhead in CWNs is minimal and specific to the particular lifting procedure chosen (e.g. the maximum ring or clique size, as already discussed above) and the way higher-order cell features are initialised (when not available). Typically, when working on molecular benchmarks, node and edge features are available and one only needs to choose an initialisation strategy for two-cells, e.g. summation or averaging of the features of lower dimensional cells. Lastly, we observed that on several benchmarks it was sufficient to adapt the hyperparameter configurations from other baselines to achieve strong (and often state-of-the-art) empirical performance.
>
> __“Is regularity important for CW Networks? Is it needed for any of the results later?”__
>
> This is an excellent question! We will make a note of this in the final version of the paper.
>
> Regularity shows up in some subtle assumptions in the proofs. However, with a more careful analysis, regularity can be relaxed in certain ways. For instance, a lifting transformation that could be useful but produces non-regular cell complexes glues the boundary of two-dimensional closed disks to all the $k$-paths in the graph. However, we preferred to maintain this property for the following reasons:
> 1. The theory of cellular sheaves, which we use in Appendix C to define the sheaf and Hodge Laplacians, relies on this assumption. In order to study cell complexes from a spectral perspective, this assumption is therefore needed.
> 2. Regularity ensures that the topology of the cell complex (i.e. the topology of the space) is completely described by its combinatorial structure (i.e. its adjacencies). So, in some sense, by considering only the adjacencies between the cells, no information about the topology of the space is lost.
> 3. Without regularity, cell complexes can become very “wild” and can be constructed from peculiar gluing procedures. Therefore, we preferred to keep things simple and clear for the reader.

---

> > ### Comment · Reviewer_hkAr · 2021-08-31
> > **thanks for your response -- and another question**
> >
> > I thank the authors for the detailed response.
> > As I revisit the theoretical results of this paper, I have some doubts regarding Theorem 15, which states that "CWL with the lifting maps from Corollary 14 is not less powerful than 3-WL."
> >
> > However, it seems that if one uses the clique complex lifting, then the following pair of graphs can be distinguished by 3-WL, but NOT by CWL with clique complex lifting:
> > G_1: consists of two 4-cycles (each component is a cycle with 4 nodes); while G_2 consists of a single component which is a 8-cycle.
> >
> > This contradicts the theorem -- or am I missing sth.?

---

> > > ### Author Response · Authors · 2021-08-31
> > > **Author response to new question**
> > >
> > > Thank you for your question! This is just a terminological confusion and we will clarify it in the final version: “Not less powerful” is _not_ the same as “more powerful” and this is why we explicitly used the negated statement.
> > >
> > > Colouring A is less powerful than B (denoted by $B \sqsubseteq A$) if any non-isomorphic pair of graphs distinguished by A is also distinguished by B. We note that this is a standard definition in the literature (for instance, see the last paragraph of Section 2 in Morris et al: https://arxiv.org/abs/1904.01543). Therefore, by logical negation “A is not less powerful than B” means that there is a pair of graphs that A can distinguish but B cannot. This is what Theorem 15 is showing. We will explicitly add the definition above in the final version to help the reader.
> > >
> > > Indeed, as the reviewer has correctly pointed out, the converse statement also holds in this case. There are pairs that 3-WL can distinguish but CWL with a clique complex lifting cannot and we will make this clear as well. CWL with its different lifting procedures effectively cuts through the k-WL hierarchy which allows it to trade-off in novel ways the computational complexity and the expressive power.

---

> > > > ### Comment · Reviewer_hkAr · 2021-09-01
> > > > **Re Theorem 15**
> > > >
> > > > Thanks for the reply!
> > > >
> > > > The use of "not less powerful" (without ever defining it) to me is ambiguous and I think the readers can easily mis-interpret it as "at least as powerful". Essentially CWL and 3-WL are simply not comparable (as neither one can distinguish all the graphs that the other one distinguish). It is like comparing apples with oranges. If the authors wish to show that there are families of graphs distinguishable by CWL but not 3-WL, then perhaps it is best to simply state as such explicitly (instead of using the ambiguous double negation "not" + "less"), followed by a remark that the same also holds for 3-WL and the two are strictly not comparable. As this is stated as a Theorem, and only one side is represented using the term "not less powerful", I feel this could be very misleading.

---

> > > > > ### Author Response · Authors · 2021-09-01
> > > > > **Author Response Re Theorem 15**
> > > > >
> > > > > We agree with the reviewer. While we adopted the terminology “not less powerful” from previous related works, we also believe the double negation may generate confusion. As already promised, we will make sure to restate the result explicitly to avoid any ambiguity.

---

### Official Review · Reviewer_rLZj · 2021-07-20

**Rating:** 4
**Confidence:** 5

**Summary:**

This paper presents an extension of the message passing simplicial network based introduced in [4] on cell complex. The authors first introduce background on regular cell complex. They present an extension of the Weisfeiler Lehman test that generalises the simplicial WL from [4] to Cellular WL and give theoretical results about the power of this new test. Then they introduce CW Networks which is a natural extension of message passing GNN to message passing on cells and show that their architecture is permutation equivariant. Finally, they show on synthetic benchmarks and real-world graphs that their architecture achieves better performances.

**Limitations And Societal Impact:**

OK

**Main Review:**

1- The idea of lifting graphs to higher order representation is interesting but not new and the authors should clarify it: [34] already showed that the expressive power is increased when dealing with high-order tensor representations of the graphs with linear layers.

2- Since Cellular WL add a lot of information, Theorem 13 is not really surprising. Indeed, the analysis made in Section 3 is rather misleading. The interest of the various WL tests is to show the trade-off between the complexity of the algorithm and its expressive power. 1-WL is less complex than 3-WL and so on but you gain in term of expressiveness.  This is the main motivation for comparing the expressive power of a GNN with the k-WL test. See [34], or Waiss Azizian, Marc Lelarge. Expressive Power of Invariant and Equivariant Graph Neural Networks, ICLR 2021, for a general overview about GNN and WL tests. In Section 3, the authors completely ignore the complexity part. For example, if I understand correctly, the clique complex lifting will include the information about all cliques in the graph which might be very costly to compute, hence Corollary 14 is just telling that if you have access to this information you can do better than a very simple algorithm. It is clearly true but not very interesting.

3- The authors mention the complexity of their GNN on page 6 and write that under some conditions the complexity is "linear in the size of the input complex". But how big can this size of the input complex be in term of the size of the graph?

4- In their Theorem 17, the authors show that CW Network layers are cell permutation invariant. Does it imply that the GNN is graph permutation invariant?

5- It seems that the authors only look at graph embedding with their CWN. Is it possible to obtain node embedding by modifying the READOUT function?

6- In the experiment section, it would have been nice to compare CIN with other algorithms having access to the same information as CIN. Since, the authors compute all the rings in the graph for their CIN, how is the perfomance of a standard GNN improved if we add the statistics of the rings in addition to the embedding computed by the GNN? Such result would allow to demonstrate whether the improvement seen in the paper is due to the new architecture or to the new features added. Similarly, you can probably increase the performance of the WL kernel by adding this new feature to get very good results.

**Time Spent Reviewing:**

4

---

> ### Author Response · Authors · 2021-08-09
> **Official response to Reviewer rLZj (1/2)**
>
> We are grateful for the time the reviewer has put into this comprehensive and actionable review, which we believe will significantly improve our paper. We have addressed below in detail each of the points that were raised.
>
> __“The idea of lifting graphs to higher order representation is interesting but not new and the authors should clarify it.”__
>
> The higher-order tensor lifting described in [34] and in _Expressive Power of Invariant and Equivariant Graph Neural Networks_ (Azizian et al.) is very different from ours and the class of functions analysed in their work does not align with the class of functions we study. Therefore, none of the results in these two papers directly apply to our work. We explain why that is the case in detail below and we will add these points in the final version of the paper.
> 1. The expressivity results from Azizian et al. rely on a lifting transformation that takes as input a graph and produces a higher-order dense tensor. The elements of this tensor are initialised with the isomorphism type of the tuple of nodes that describe it (see the last paragraph of Section 2.2 in https://arxiv.org/pdf/2006.15646.pdf). In contrast, our lifting map produces a cell complex. If the cell complex is regular, then indeed it can be represented by a tensor. However, this representation is a sparse tensor with certain sparsity constraints that yield a valid (regular) cell complex. More formally, this is called an incidence tensor or incidence structure / geometry (see https://arxiv.org/abs/1905.11460). To the best of our knowledge, no one studied lifting transformations producing incidence tensors and no one studied the expressive power of models operating on such tensors.
> 2. To see why sparsity is important, to represent k-rings, one needs a tensor of order $k$ (i.e. with $n^k$ entries, where $n$ is the number of nodes). For almost all experiments, we use $k \geq 6$ and a dense tensor of this size would be too large for any practical purpose. Therefore, the models from Maron et al. cannot explicitly model rings of typical sizes for practical molecular applications because of the computational and memory complexity. By exploiting the sparsity of the cell complex (both its higher-order structures and cell adjacencies) we are able to achieve an effective tradeoff between expressive power and computational efficiency (more details about this in the next bullet point).
> 3. The class of functions we study is also very different from the class of functions studied in Maron et al. and Azizian et al. The mentioned papers study functions composed of all the linear node permutation equivariant/invariant functions operating on dense tensors describing the graph. In contrast, our paper studies a class of message passing functions that are cell permutation equivariant operating on cell complexes. Not only is the input space of these functions different but also the symmetries they have are different. Lastly, differently from these methods, we retain the inductive bias of “locality” which characterises neural message passing and has so far significantly contributed to the success of graph neural models.
> 4. Note that a sparse tensor with specific sparsity constraints is only one possible representation for the cell complex. In the paper, we consider the more common representation based on boundary operators (see Definition 28 in the Appendix B), inherited from algebraic topology and differential geometry. Cell complexes inherit many connections with these fields (some of which we discuss and exploit in the appendix). In contrast, the tensors from Maron et al. and Azizian et al. represent regular dense tensors.
>
> __Tradeoff between expressive power and computational complexity. “The authors completely ignore the complexity part.”__
>
> We understand that there is a fundamental tradeoff between expressivity of the isomorphism test and its computational complexity. One of the main features of the proposed model is exactly to balance these two desiderata and we will strive to better convey this in the paper. As mentioned above, by exploiting the sparsity of cell complexes, we produce efficient and theoretically expressive models for molecular graph problems:
> 1. It is not necessary to find all the cliques or rings in the graph. We acknowledge to have stated the theorems in the main text in an oversimplified manner and that seems to have generated some confusion. We will restate them more precisely in the final version. The discussed clique complex and induced cycle lifting transformations have an associated parameter $k$ specifying the size of the maximum clique or induced cycle, respectively. As the proofs show, the WL expressivity result (Corollary 14) holds for any $k \geq 3$ and the 3-WL result (Theorem 15) holds for any $k \geq 4$. In molecular graphs, which are topologically planar, 3-rings / cliques can be found in linear time in the size of the graph, while 4-rings can be found in $O(n)$ and listed in $O(n^2)$ (see the classic result from Chiba and Nishizeki: http://www.cs.cornell.edu/courses/cs6241/2019sp/readings/Chiba-1985-arboricity.pdf ). Therefore, our model is strictly more expressive than WL with no additional computational burden (i.e. linear complexity in the size of the graph) when considering only molecular graphs. In this case, it is also not less powerful than 3-WL with only $O(n^2)$ complexity.
> 2. More generally, the number of rings of size strictly greater than $4$ is also empirically contained for molecular graphs and is typically upper-bounded by a small constant (see next paragraph). This is also backed up by our wall-clock time experiments (see Appendix A.5 which also contains a detailed computational analysis) where a GIN model with a similar number of parameters is only between 19% and 35% faster than our model at inference time on the ZINC dataset.
> 3. Even in general graphs, 3-rings / cliques can be efficiently found in $O(m^{3/2})$, where $m$ is the number of edges. Therefore the WL result is also satisfied in general graphs without requiring the higher complexity of 3-WL based approaches.
> 4. In agreement with the theory, in all the real-world graph experiments we conducted, we employed a finite $k$ for the maximum clique or ring size, parameter mentioned for every benchmark. Therefore, it is not necessary to find all such structures in the graph.
>
> We hope this clarifies that our model does indeed balance the tradeoffs between computational complexity and expressive power.
>
> __“[...] how big can the input complex be in terms of the size of the graph?”__
>
> This is comprehensively discussed in Appendix A.5. In general, the size of the complex is given by the overall number of cells, at each dimension. For all practical purposes we consider two-cell complexes, that is complexes with cells of maximum dimension two. For skeleton-preserving liftings, the number of 0- and 1-cells always corresponds, respectively, to the number of nodes and edges (i.e. the size of the graph - let us denote it by $S$). The additional complexity may therefore be dictated by the number of two-cells (e.g. rings), which depend on the lifting procedure. For rings, this can be exponential in general. However, for molecular modelling, the number of rings can be upper bounded by some small constant $C$, and the complexity of the message passing procedure becomes $O(S + C)$.
>
> Experimentally, we observed the overall number of rings per molecule to be, indeed, contained. We counted rings up to 12 nodes on _ZINC-full_ and _MOLHIV_: each molecule has an average number of $\sim 2.78$ rings on the former and $\sim 3.0$ on the latter. Additionally, the maximum recorded number of 6-rings (the most represented ones) in _ZINC-full_ is $7$, while it is $32$ on _MOLHIV_. These statistics empirically verify that, in molecules, the number of these substructures is far from exponential.

---

> ### Author Response · Authors · 2021-08-09
> **Official response to Reviewer rLZj (2/2)**
>
> __“The authors show that CW Network layers are cell permutation invariant. Does it imply that the GNN is graph permutation invariant?”__
>
> We remark that, in Theorem 17, we show that CWN layers are permutation _equivariant_ (as opposed to _invariant_). It is trivial to show that an architecture constructed by stacking these layers is also equivariant and that, when a cell permutation invariant readout function is used, the overall model is cell-permutation invariant. When the model $f_\theta$ is composed with a lifting transformation $g$, the function $f_\theta \circ g$ is node permutation invariant.
>
>
> __“It seems that the authors only look at graph embedding with their CWN. Is it possible to obtain node embedding by modifying the READOUT function?”__
>
> A READOUT function is typically applied after a series of message passing layers whenever it is required to produce a single representation for the overall complex / graph by aggregating the representations of all cells / nodes. To obtain node embeddings no such aggregation is required, and it is natural to directly consider the representations of 0-cells at the last layer of the architecture, which are in 1-to-1 correspondence with nodes in the original graphs. However, since the expressivity results are particularly relevant for graph classification, we did not consider node-level tasks in this work.
>
> __“It would have been nice to compare CIN with other algorithms having access to the same information as CIN.”__
>
> We would like to stress that we have already compared CIN with two such strong baselines: GSN (ref. [9]) and HIMP (ref. [26]).
> When applied to molecular tasks, GSN augments the features of the nodes with the number of (induced cycles) they are part of, before running regular message passing. HIMP directly models the presence of rings by building an auxiliary graph hierarchy based on the junction tree representation of the input molecule. Although this hierarchy allows us to also model adjacencies between rings, edges and nodes, it is different from the cell complex hierarchy in our work. Both of these methods represent strong molecular baselines, but are outperformed by CIN in all our large molecular benchmarks; GSN is also often outperformed in the smaller TU datasets. We will make sure to stress the importance of these baselines in the paper, and the considerations that led to their selection.
>
> We would also like to mention the ablation study on ZINC reported in Appendix D.4. Here we clearly identify the source of the improvements as the inclusion of rings in the cellular message passing procedure. We will refer to this ablation study in the main text.

---

> > ### Comment · Reviewer_rLZj · 2021-08-19
> > **after rebuttal**
> >
> > Thank you for your answer. I understand that sparsity of cell complexes is essential for your approach and I advise you to include the elements provided in your response as well as the references in the main part of your paper. I still maintain that the presentation itself is not strong enough for acceptance, and not something that is possible to fix before camera-ready.
> >
> > Minor comment, I still do not understand in Th 17 if cell permutation invariant is the same as node permutation invariant.

---

> > > ### Author Response · Authors · 2021-08-20
> > > **Post-rebuttal official response to Reviewer rLZj**
> > >
> > > We respectfully disagree! We believe we have addressed all the points raised by the reviewer and, in our previous response, we mostly emphasised aspects that are already included in the paper and might only require minor clarifications:
> > >
> > > 1. The computational complexity of the model and the size of the cell complex are discussed in great detail in Appendix A.5. These analyses are also backed up by detailed wall-clock time experiments in the same section. Taken together, they clearly demonstrate the computational efficiency of the method for the tasks of interest in this work. We will also strive to provide a better overview of A.5 in the main text.
> > > 2. We have already compared our method against strong baselines having access to the same information as CIN (i.e. GSN and HIMP) and our method clearly outperforms them. We also already included an ablation study in Appendix D.5 that clearly identifies the presence of the rings in the message passing procedure as the main source of improvement.
> > > 3. The mentioned related work from [34] is already included as a baseline in our TU dataset experiments and our model consistently outperforms it. As promised, we will also include a discussion about the recent results from Azizian et al, which, as explained above, do not overlap with ours.
> > >
> > > Therefore, it is unclear what aspects of the presentation the reviewer still finds “not strong enough for acceptance”. Furthermore, the claim that this is “not something possible to fix before camera-ready” is beyond the range of subjective discretion of the reviewer.
> > >
> > > Finally, to answer the last comment, cell permutation equivariance / invariance is, in general, **_not_** the same as node permutation equivariance / invariance. Nodes are zero-dimensional cells. In contrast to GNNs, CWNs treat cells of different dimensions (e.g. vertices, edges, 2-cells, etc.) as independent objects with independent feature matrices that can be independently permuted by different permutation operators together with the linear boundary operators that relate them. CWNs are equivariant / invariant with respect to these more general transformations. This is discussed in detail in Appendix B.

---

### Official Review · Reviewer_23iC · 2021-07-28

**Rating:** 8
**Confidence:** 3

**Summary:**

The paper focuses on generalizing graph neural networks to model higher-order structures. It proposes a new message passing scheme operating on regular cell complexes. The induced models, called CW Networks, generalize and subsume the message passing simplicial networks (MPSNs) which operate on simplicial complexes, and are proved more powerful than the WL test and in some cases not less powerful than the 3-WL test.




**Limitations And Societal Impact:**

No major negative social impact.

**Main Review:**

The paper is well written. It makes a comprehensive theoretic analysis of the proposed model as well as a thorough empirical study. The concept of using cell complexes seems to be a natural but novel extension of simplicial complexes and has some intuitive advantages on molecular datasets (with a lot of rings). The experimental results are also quite good compared to other recent models. I think it is already in good shape and ready for publication.

As to the negative points, the authors actually already indicated the limitation of the method in the paper, i.e. the high computational complexity on more general large graphs, which would impede the applicability of the model.

Another minor problem may be the experiments of long-range interaction. The synthetic data only contains chordless rings and it seems all nodes are in the same rings. That may be not a general case, or we can say it just covers one case of long-range interactions. To make the experiments more convincing, maybe in addition to the central rings there should also be some associated branches?

I have also another question for the authors: if we pre-process the graphs to decompose them into nodes, edges, rings, and use these elements as nodes to form one heterogenous graph, would a GNN on that graph be equal to the cellular message passing to some extent?



**Time Spent Reviewing:**

5

---

> ### Author Response · Authors · 2021-08-09
> **Official response to Reviewer 23iC**
>
> We would like to thank the reviewer for the time they have put into this thorough review. We are glad to see the reviewer found our empirical and theoretical analysis comprehensive and thorough. The remaining comments and questions are addressed below.
>
> __High computational complexity__
>
> Although the number of higher-order cells may grow exponentially in general graphs, it is contained for all the tasks of interest in this work. In molecular modelling, where the graphs are (almost always) planar, the number of rings is empirically contained: we found each molecule to contain on average $\sim 2.78$ rings with no more than 12 nodes on _ZINC-full_ and $\sim 3.0$ such rings on _MOLHIV_ (larger rings are significantly underrepresented in these datasets).
>
> Additionally, it is possible to restrict the lifting procedure to substructures of a limited size based on domain knowledge (if available) or statistics computed on a dataset sample. For instance, 3-rings (triangles) can be listed in linear time and 4-rings in quadratic time in planar graphs (including molecules). We stress once more that one may also resort to bio-molecular libraries that are specialised in detecting such structures, such as _RD-kit_.
>
> Even for general graphs, listing 3-rings and 4-rings can be shown to depend on the arboricity of the graph (a measure of graph sparsity – see http://www.cs.cornell.edu/courses/cs6241/2019sp/readings/Chiba-1985-arboricity.pdf). So, the sparser the graph, the faster such structures can be detected. We remark that 3-rings are sufficient to make CWNs strictly more powerful than the WL test.
>
> __“Another minor problem may be the experiments of long-range interaction. [...] Maybe in addition to the central rings there should also be some associated branches?”__
>
> We confirm that samples all consist of one chordless cycle only, and that ‘source’ and ‘target’ nodes are part of this cycle. We agree on the fact that this is not generally the case for real-world graphs, which may contain more than one ring and, usually, additional structures. However, the aim of the experiment was to verify one specific hypothesis: that our model is able to learn an efficient message passing scheme to capture long-range interactions across rings thanks to the additional adjacencies induced by two-cells. Thus, the need for a controlled and simplified setting. This setting has also practical importance because some molecular properties depend on atoms placed on the opposite side of a ring (see http://proceedings.mlr.press/v70/gilmer17a.html, https://www.nature.com/articles/sdata201422).
>
> We also note that the shortcuts created by the rings are not limited to the atoms of the same ring but may produce shortcuts between any pairs of nodes in the graph by compressing the minimum-length path between them. This effect is particularly pronounced if the two nodes are part of a long chain of adjacent rings. Experiments on ogbg-molhiv, where the model obtains SOTA results with only two layers, additionally provide empirical confirmation. This should be compared with other high-performing baselines on this dataset that use more layers such as DeeperGCNs (https://arxiv.org/pdf/2006.07739.pdf) and millions of parameters, such as Graphormers (https://arxiv.org/pdf/2106.05234.pdf).
>
>
> __“If we pre-process the graphs to decompose them into nodes, edges, rings, and use these elements as nodes to form one heterogenous graph, would a GNN on that graph be equal to the cellular message passing to some extent?”__
>
> A standard GNN model on a graph where the original nodes, edges and rings constitute its (heterogeneous) nodes would not equate cellular message passing on the ring-lifted graph.
>
> Our architecture distinguishes between the different neighboring relations (boundary cells, upper-adjacent cells, etc.); each cell receives a separate aggregated message for each of them. CWNs feature message functions specific to each of these adjacencies and at each cell dimension. The described GNN, on the contrary, would not make any distinction between different neighboring relations and one single generic message would always be exchanged between nodes, disregarding the nature of the adjacency. Lastly, the described graph would not be able to equivalently describe, for example, the upper adjacencies between edges part of the same ring of size $N$. This would require a “hyperedge” formed by the $N$ edges forming the ring. Contrary to the described GNN, in a CWN each of these adjacencies generates a message, and (importantly) this would include the features of the shared ring which induced the neighboring relation.

---

### Author Response · Authors · 2021-08-09
**Official overall response**

We would like to thank the reviewers for their extensive reviews, helpful comments and actionable suggestions for improving our manuscript. We are of course delighted that the reviewers generally view our work favourably, and that they deem our paper “novel” (_23iC_, _1iwy_), “useful in practical applications” (_hkAr_), with a “solid theoretical analysis” (_1iwy_). They also appreciated our empirical results (_23iC_, _rLZj_, _hkAr_, _1iwy_). We are also grateful they have focused on concrete suggestions for improving the manuscript, particularly in the presentation of technical details, which we will be able to act upon. Furthermore, we appreciate the insightful questions that will help improve the clarity of our discussion.

We have responded to each review directly to address the specific comments they make. Again, we would like to express our gratitude for these constructive reviews and welcome further comments or requests.

---

### Decision · Program_Chairs · 2021-09-27

**Decision:**

Accept (Poster)

**Comment:**

This paper proposes a new graph neural network framework allowing message passing for high-order structures. The paper provides clear theoretical foundation and also delivers convincing empirical evidence. Although it was mentioned that the contribution is a bit incremental over existing work [4,24], overall the paper was positively/passionately accepted. One reviewer did raise strong concern about the presentation of the paper. But based on the paper and all other reviewers' opinions, the presentation concern does not seem general. Therefore, the AC recommends the paper to be accepted.